# Judging the difficulty of perceptual decisions

Anne Löffler[1,2,3], Ariel Zylberberg[1,2†], Michael N Shadlen[1,2,3,4†], Daniel M Wolpert[1,2*†]

[1]Zuckerman Mind Brain Behavior Institute, Columbia University, New York, United States; [2]Department of Neuroscience, Columbia University, New York, United States; [3]Kavli Institute for Brain Science, Columbia University, New York, United States; [4]Howard Hughes Medical Institute, Columbia University, New York, United States

*For correspondence: wolpert@columbia.edu

†These authors contributed equally to this work

Competing interest: The authors declare that no competing interests exist.

**Abstract** Deciding how difficult it is going to be to perform a task allows us to choose between tasks, allocate appropriate resources, and predict future performance. To be useful for planning, difficulty judgments should not require completion of the task. Here, we examine the processes underlying difficulty judgments in a perceptual decision-making task. Participants viewed two patches of dynamic random dots, which were colored blue or yellow stochastically on each appearance. Stimulus coherence (the probability, $p_{blue}$, of a dot being blue) varied across trials and patches thus establishing difficulty, $|p_{blue} - 0.5|$. Participants were asked to indicate for which patch it would be easier to decide the dominant color. Accuracy in difficulty decisions improved with the difference in the stimulus difficulties, whereas the reaction times were not determined solely by this quantity. For example, when the patches shared the same difficulty, reaction times were shorter for easier stimuli. A comparison of several models of difficulty judgment suggested that participants compare the absolute accumulated evidence from each stimulus and terminate their decision when they differed by a set amount. The model predicts that when the dominant color of each stimulus is known, reaction times should depend only on the difference in difficulty, which we confirm empirically. We also show that this model is preferred to one that compares the confidence one would have in making each decision. The results extend evidence accumulation models, used to explain choice, reaction time, and confidence to prospective judgments of difficulty.

## eLife assessment

This study investigates how humans make decisions on the difficulty of perceptual categorization tasks. The study finds that such judgments are best described by an evidence-accumulation model that includes a dynamic comparison of difficulty-related evidence, which terminates when the difference in evidence between two tasks reaches a predetermined bound - a **valuable** finding for research in perceptual decision-making. The paper provides **compelling** behavioral evidence for the proposed model through: 1/ quantitative model selection/validation procedures, and 2/ qualitative analyses of the relation between the optimal model of the task and the human data (and the proposed model).

## Introduction

Estimating the difficulty of different tasks we might perform allows us to decide which task to engage in *Bennett-Pierre et al., 2018*; *Wisniewski et al., 2015*, allocate the necessary amount of effort or cognitive control (*Shenhav et al., 2013*; *Dunn et al., 2019*), and predict our ability to perform the task successfully (*Morgan et al., 2014*; *Siedlecka et al., 2016*; *Fleming et al., 2016*, *Moskowitz*

et al., 2020). The need for difficulty estimation arises in a wide variety of human endeavors, from attempting different recipes and hiking routes to learning different languages. For example, consider a musician who is deciding which piece to learn to improve her skills. If the piece is too easy, she is likely to be bored, while if the piece is too difficult she may be frustrated, learning very little in either case. To make the correct choice, she must accurately estimate the difficulty of each potential piece given her current abilities. Attempting to learn each piece would lead to accurate estimates of their difficulty, yet this defies the purpose, since the difficulty estimation was sought to decide if the piece was worth learning in the first place. Alternatively, she could allocate a fixed period of time or learn the first few bars of each piece, or use cues like the length, key or tempo of each piece to estimate their difficulty.

Past studies of difficulty judgments have mainly relied on complex tasks, like five-grade students judging the difficulty of remembering a sentence (Stein et al., 1982) or physics teachers evaluating the difficulty of different exam questions (Fakcharoenphol et al., 2015). This line of research highlights the important role that cues and heuristics play in estimating task difficulty (Vangsness and Young, 2019). While the use of complex naturalistic tasks ensures that difficulty estimates are realistic, their complexity precludes quantitative modeling and thus the identification of the inference process underlying the formation of difficulty judgments.

Tasks that require consideration of multiple samples of evidence presented sequentially over time have advanced our understanding of the computational and neurophysiological underpinnings of decision making. In such tasks, the signal-to-noise ratio of the samples is varied across trials, and on each trial we can obtain the objective measures of choice (correct vs. incorrect) and reaction time (RT). However, metacognitive judgments about the decision process itself, such as confidence and difficulty, are also of interest. Decision confidence is often defined as the probability that a choice, just made, is correct or appropriate (Peirce and Jastrow, 1884; Kiani and Shadlen, 2009). Similarly, difficulty can be considered a metacognitive judgment, like confidence. In fact, confidence in a decision one has made can be converted to difficulty (the signal-to-noise ratio) if one knows how long it took to reach the decision (Kiani et al., 2014). In this case confidence and difficulty are retrospective metacognitive judgments that emphasize, respectively, the probability of having chosen correctly and the effort that was required.

Difficulty estimation can be prospective or retrospective, and can even be independent of task performance (Desender et al., 2017). For example, the musician may learn a piece and then judge its difficulty (a retrospective estimate), use her prior knowledge about the composer (a prospective estimate), or learn the first few bars and extrapolate its difficulty to the rest of the piece (both prospective and retrospective). Our interest is in prospective judgments of difficulty, for which confidence in the outcome of the decision can be viewed only as a prediction, governed by an assessment of difficulty and introspection about one's own acumen.

Here, we address how a subjective sense of difficulty is constructed from a sequence of evidence samples obtained from the environment, and how this difficulty decision is terminated. In our main experiment, human participants were presented with two visual stimuli (two flickering blue and yellow patches of dots) and had to report for which one it would be less difficult to make a decision about the dominant color. Our results show that judgments of difficulty obey a sequential sampling process similar to one that would be used to make a single decision about perceptual category. However, rather than accumulating evidence for color, difficulty judgments rely on tracking the difference in the absolute accumulated evidence for color judgments for the two stimuli.

## Results

Participants performed variants of a perceptual task that required binary decisions about either one or two patches of dynamic random dot displays. In some blocks they reported the dominant color of one patch (color judgment) or which of the two patches it would be easier to make a color judgment about (difficulty judgment). The task was performed as a response time version (Experiment 1). We then evaluate models that explain the difficulty judgments and RTs. The best model makes a prediction, that we test in a second experiment.

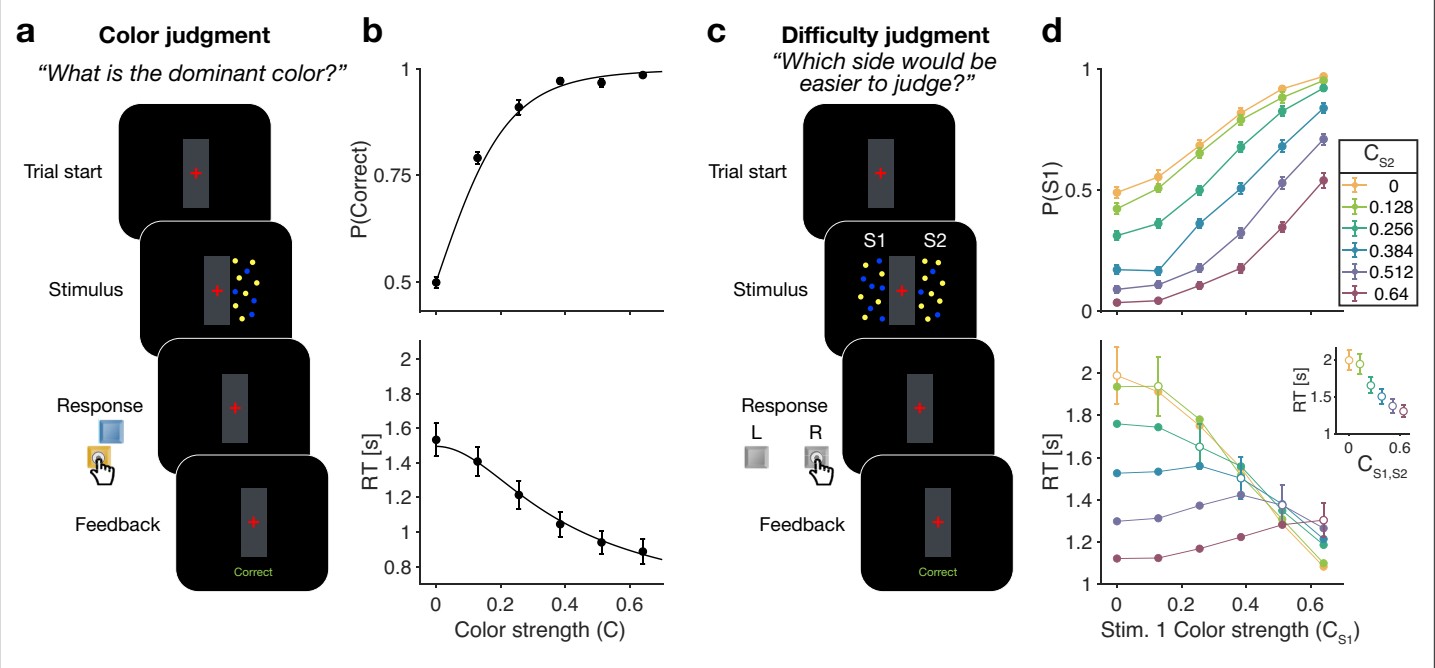

**Figure 1.** Reaction time color and difficulty judgment tasks (Experiment 1). (**a**) Schematic of the color judgment task. Participants judged the dominant color of a single random dot stimulus consisting of blue and yellow dots. (**b**) Proportion of correct choices (*top*) and reaction time (*bottom*) as a function of color strength. Solid lines show the average of fits of a standard drift diffusion model to each participant's choices and reaction times (RTs). Values of the fit parameters are shown in *Supplementary file 1*. Data points show mean ± 1 SEM from 20 participants. (**c**) Schematic of the difficulty judgment task. Participants decided for which of the two stimuli it was easier to judge the dominant color, regardless of whether that stimulus was blue or yellow dominant. (**d**) Proportion of trials in which stimulus 1 (**S1**) was chosen as the easier stimulus (top) and RTs (bottom) as a function of the strength of S1 (abscissa) and S2 (colors). Open circles identify conditions where both patches have the same color strength (also plotted in inset). Data points show mean ± 1 SEM from 20 participants.

## Color judgments in an RT task (Experiment 1a)

Participants were first exposed to a color judgment task in which they were asked to decide whether the dominant color of a patch of dynamic random dots was yellow or blue (*Figure 1a*) over a range of difficulties (*Mante et al., 2013*; *Bakkour et al., 2019*; *Kang et al., 2021*). The difficulty of the color choice was conferred by the probability that a dot would be colored blue ($p_{\text{blue}}$) or yellow ($1 - p_{\text{blue}}$) on each frame. Therefore, the signed quantity $C^{\pm} = 2(p_{\text{blue}} - 0.5)$, termed the color coherence, takes on positive values for blue dominant stimuli. We refer to the unsigned quantity $C = |C^{\pm}|$ as color strength and this took on six different levels {0, 0.128, 0.256, 0.384, 0.512, 0.64}. The color strength determines the difficulty of the task with smaller strengths being more difficult. This task served to familiarize the participant with the task and instill the intuition that some decisions are more difficult than others. Only participants who performed above a set criterion (see Materials and methods) were invited to perform the main experiment. We refer to this task as a color judgment.

*Figure 1b* shows choice accuracy and RTs averaged across ($N = 20$) participants for color judgments. Participants' choices and RTs varied according to color strength. As expected, participants chose the correct color more often, and they responded faster, when the strength increased. The data are well described by a standard drift diffusion model (black lines) that explains the choice and RT by applying a stopping bound to the accumulation of the noisy color evidence (see Materials and methods). We refer to the accumulation of color evidence (blue minus yellow) as the color decision variable, $DV$.

## Difficulty judgments in an RT task (Experiment 1b)

The main experiment required participants to make difficulty judgments (*Figure 1c*) of two stimuli simultaneously presented to the left and right of a central fixation cross. Participants had to select the stimulus for which they thought it would be easier to decide what the dominant color would be,

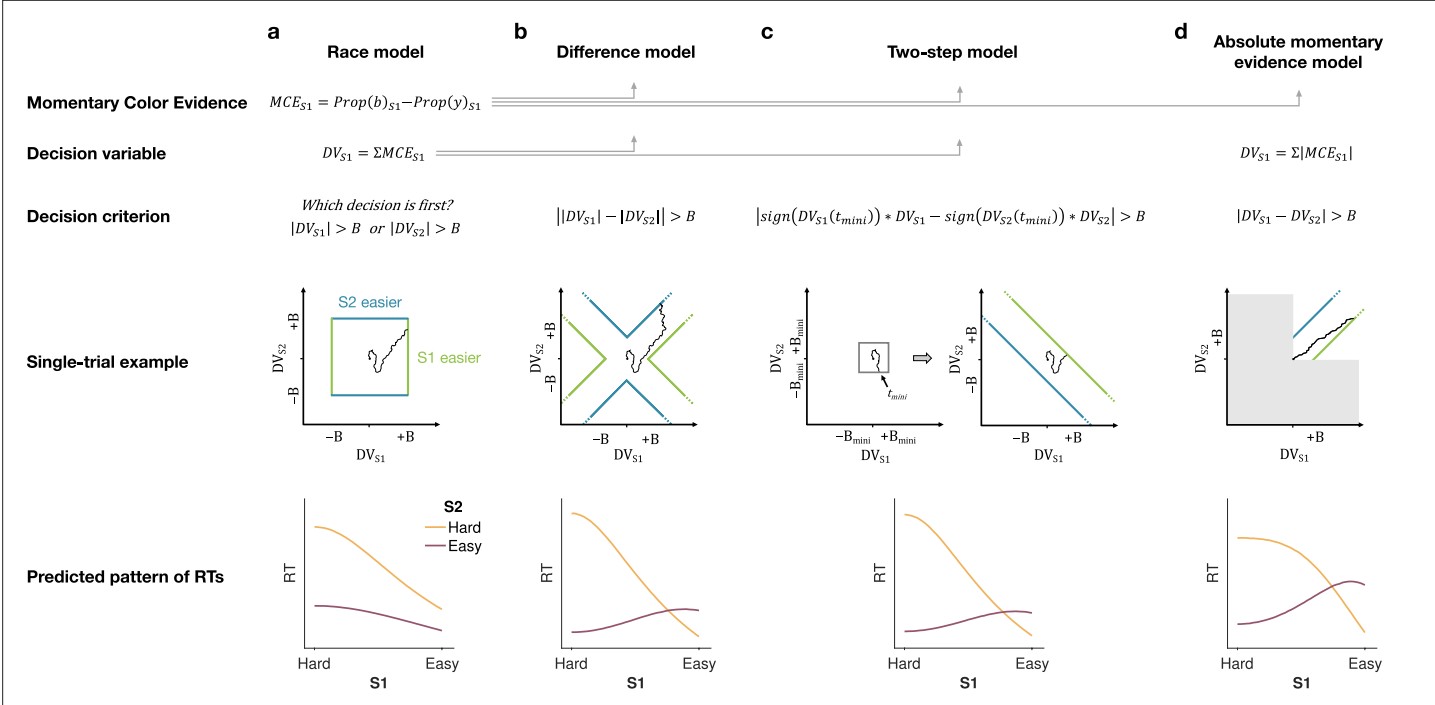

**Figure 2.** Models of difficulty judgments. In each model, momentary color evidence (MCE) is obtained simultaneously from each of the two stimuli as the difference in proportion of blue vs. yellow dots on each frame. The models differ in (i) how this momentary evidence is accumulated into a decision variable (second row) and (ii) how the bounds are set for the decision (third row). For each model, an example of a simulation of a single trial (fourth row) is illustrated in the 2D space of the decision variables $DV_{S1}$ and $DV_{S2}$ for the left (**S1**) and right (**S2**) stimuli. Each simulation shows a biased random walk that starts at the origin and terminates when it reaches one of the decision bounds. Decision bounds in green and blue correspond to S1 and S2 being judged the easier decision, respectively. For clarity, all bounds are illustrated as time-independent (although in the model they are allowed to collapse over time). The models predict different patterns of reaction times (RTs; bottom row) for different difficulty combinations of S1 (abscissa) and S2 (magenta = easy, yellow = hard). (**a**) The race model, (**b**) the difference model, (**c**) the two-step model, and (**d**) the absolute momentary evidence model (gray area is not reachable as all accumulation is positive). See main text for model details.

regardless of whether the patch was yellow or blue dominant. Importantly, in the difficulty judgment task, participants were never asked to report color decisions for either stimulus. In this task all 12 × 12 coherence combinations of the two stimuli were presented in a randomized order (see Materials and methods). We refer to this task as a difficulty judgment.

In the difficulty task, choice and RTs were affected by both the left (S1) and right (S2) stimulus strengths (*Figure 1d*). Participants chose S1 more often with increasing strength of S1 (*Figure 1d* top, abscissa) and decreasing strength of S2, and vice versa. The RTs exhibit a more complex pattern. For a given S1 strength (*Figure 1d* bottom, abscissa), RTs tend to be slowest when the strength of both stimuli are the same (open symbols), and the RTs accompanying these same matched difficulties are shorter for higher strength (*Figure 1d* bottom, inset). Therefore, the RTs are not simply a function of the difference between the two strengths. For example, RTs were significantly longer in the hard-hard (0:0, leftmost point in inset) combination (mean =1.99 s, sd =0.6) compared to easy-easy (0.64:0.64, rightmost point in inset) conditions (mean 1.30 s, sd =0.36, difference in RT $t(19) = 8.63$, $p < 0.001$, $CI[0.52 − 0.85]$). These tendencies lead to a criss-cross pattern in the RTs. For example, when S1 is difficult the shortest RT occurs when S2 is easy, whereas when S1 is easy, the longest RT occurs when S2 is easy. This leads to an inversion in the order of colors in the leftmost and rightmost stacks of points.

We developed four models (*Figure 2*) of difficulty decisions and examined whether they could account for the choice and RT data. The models we consider assume that sensory evidence accumulates over time. This is a reasonable assumption because these models have been very successful in explaining choice, RT, and even confidence in many perceptual and mnemonic decision tasks, including color discrimination (*Bakkour et al., 2019*; *Gold and Shadlen, 2007*). All models assume that for each stimulus (S1 and S2), momentary color evidence (MCE) for blue vs. yellow (positive for

blue) is extracted at each point in time. The models differ in how they use this MCE to determine which stimulus is easier.

## Race model

In the *race model*, the difficulty decision is determined by which of two color decisions terminates first, motivated by the regularity that more difficult stimuli typically require more samples of evidence.

In this model, MCE is accumulated into two independent DVs, $DV_{S1}$ and $DV_{S2}$, which represent the accumulated color evidence for the left and right stimulus, respectively. The single-trial example in *Figure 2a* shows a schematic of the decision process, plotted as $DV_{S2}$ against $DV_{S1}$. In this space the bounds form a square at $\pm B$ in each dimension. The DVs both start at zero so that the trajectory starts at the origin and evolves as a function of time. The time dimension is not shown, so the displayed trajectory is the path of the DVs up until the decision is made. The trajectory is confined to a space bounded by the green and blue termination bounds. The bounds are actually collapsing as a function of time but form a square at any point in time. Trajectories that reach the green or blue bound lead to S1 or S2 being judged as easier, respectively. In the example shown, S1 is judged as easier.

The bottom row of *Figure 2a* shows the predicted pattern of RTs. When both color decisions are hard, difficulty choices are slow (long RT) since they depend on the winner of two slow processes. RTs decrease as either of the two color decisions becomes easier. When both color decisions are the easiest (0.64:0.64), RTs should be fastest since they reflect the minimum of two fast processes. This model, therefore, predicts that the shortest RTs occur when both stimuli are easiest, which is clearly contradicted by the data (e.g., the rightmost stack of points in *Figure 1d*, bottom).

## Difference model

In the *difference model* (*Figure 2b*), difficulty is determined by the difference between the absolute values of the DVs, motivated by the intuition that the accumulation of weaker evidence is unlikely to traverse far from the origin compared to the accumulation of stronger evidence.

In this model the difficulty decision is made by computing $|DV_{S1}| - |DV_{S2}|$. When this difference value reaches an upper/lower bound, then S1/S2 is chosen as the easier one. Thus, in contrast to the race model, the decision boundary is applied to $|DV_{S1}| - |DV_{S2}|$, instead of the individual $DV$s. This gives rise to bounds that parallel the positive and negative diagonals on which $|DV_{S1}| - |DV_{S2}| = 0$. This leads to bounds that form channels emanating from the origin. Note that the four apices on the bounds are at $(0, \pm B)$ and $(\pm B, 0)$, which correspond to where a decision would be made, that is $|DV_{S1}| - |DV_{S2}| = \pm B$. Similarly, all other points on the bounds correspond to this difference being equal to $\pm B$. Again, trajectories that reach the green or blue bound lead to S1 or S2 being judged as easier, respectively. In this example S2 is judged as easier.

The difference model predicts a criss-cross pattern of RTs that is qualitatively similar to the data. In the difference model the RT is related to the difference in the absolute coherences of each stimulus. In general, the larger the unsigned difference the shorter the RT. Therefore on the left of the RT graph, where one of the stimuli (abscissa) is hard, RT will be longer when the other stimulus is also hard (yellow, same coherence) compared to when it is easy (purple, large coherence difference). In contrast, on the right-hand side of the RT graph where one stimulus is easy (abscissa) the RT will now be longer when the other stimulus is also easy (purple, no coherence difference) and shorter when the other coherence is harder (yellow). This leads to the criss-cross observed in the simulation.

The model also predicts that the easy-easy comparison takes less time than the hard-hard comparison (right end of purple compared to left end of gold). We will supply an intuition for this prediction under Experiment 2, which is designed to test it.

## Two-step model

The *two-step model* combines elements of the first two models and involves a two-step process. The motivation here is that if the dominant color of each stimulus were known, the difficulty decision would be simpler to compute and more accurate. Therefore, this model operates in two steps, the first of which estimates the dominant color of each stimulus and the second judges difficulty based on these estimates.

The two-step model (*Figure 2c*) first estimates the signs of the two strengths with a mini-color decision (made at time $t_{mini}$). Similar to the race model, the mini-decision terminates as soon as one

of the DVs has reached a bound. In the second step, further evidence is accumulated and a difficulty decision is made when

$$\text{sign}\left[DV_{S1}(t_{mini})\right] \cdot DV_{S1} - \text{sign}\left[DV_{S2}(t_{mini})\right] \cdot DV_{S2} = \pm B.$$

For example, if S1 is estimated to have a positive coherence and S2 a negative coherence the decision should be made when $DV_{S1} + DV_{S2} = \pm B$, whereas if both coherences are estimated to be positive the decision should be made when $DV_{S1} - DV_{S2} = \pm B$. Note that if no further information is accumulated after the mini-decision, then the two-step model becomes identical to the race model. Differences in predictions between the difference and two-step arise from trials in which there is a mismatch between the inferred dominant colors from the two-step model and the color associated with the final DVs in the difference model.

This model predicts a similar RT pattern to the difference model, but it requires one additional parameter: a decision bound for the initial color choice $B_{mini}$. The higher the bound, the longer the initial color decision will take, but the more accurate the estimate of the sign of each coherence will be. When both stimuli are hard, the initial color choice will be slow, thus causing overall longer RTs for hard-hard compared to easy-easy conditions. Additionally, similar to the difference model, the duration of the second step depends on the difference in difficulty. When both stimuli are equally hard (or easy), the difference in DVs will take longer to reach a bound compared to conditions where the difference in difficulty is large. Thus, similar to the difference model, this model predicts a crossing in the pattern of RTs where decisions about easy-easy stimuli take longer than decisions about easy-hard stimuli.

### Absolute momentary evidence model

In the *absolute momentary evidence model*, each DV represents the accumulated *absolute* MCE, as opposed to the signed evidence used in the first three models. The motivation here is the simplicity of accumulating momentary evidence bearing directly on difficulty. The obvious shortcoming is that this intuition is only valid if the samples have the same sign as the color coherence; in other words it ignores the contribution of noise.

In the model (*Figure 2d*), on each frame absolute color strength is derived from each stimulus (independent of the color dominance of each frame) and the difference between these absolute color strength is accumulated over frames. A difficulty decision is made when the difference between the DVs reaches a bound.

Compared to the difference model, the absolute momentary evidence model generally predicts faster RTs for hard-hard conditions. In the difference model weak evidence for blue followed by weak evidence for yellow for one stimulus cancel each other out in terms of the DV for that stimulus. In contrast, in the absolute momentary evidence model as the evidence is summed unsigned, such weak evidence for each color causes an increase in DV and in general contributes to the accumulated differences between the two stimuli.

## Model fitting

In each model, there are two separate streams of sensory evidence—one for each stimulus. For simplicity, we illustrate the models in *Figure 2* as if the two streams of evidence were accumulated in parallel. However, in a previous study, we showed that decisions about two perceptual features (or two stimuli) are based on serial, time-multiplexed integration of decision evidence, causing longer RTs for double decisions compared to decisions about a single stimulus (*Kang et al., 2021*). We include this feature in our model, but its only consequence is to expand the decision times, without affecting distinguishing features of the models (see Materials and methods for additional explanation).

We fit all four models to each participant's difficulty choices and RTs (see Materials and methods and Model recovery). For each model, five parameters were optimized: a drift rate $\kappa$ that relates coherence to the mean rate of accumulation of color evidence, three parameters controlling how each decision bound collapses over time, and a non-decision time $T_{nd}$ reflecting sensory and motor processing time that add to the decision time to yield the measured RT. For the two-step model, one additional parameter $B_{mini}$ was required for the bound height of the initial color choice. We fit the model by maximum likelihood to each participant's data individually.

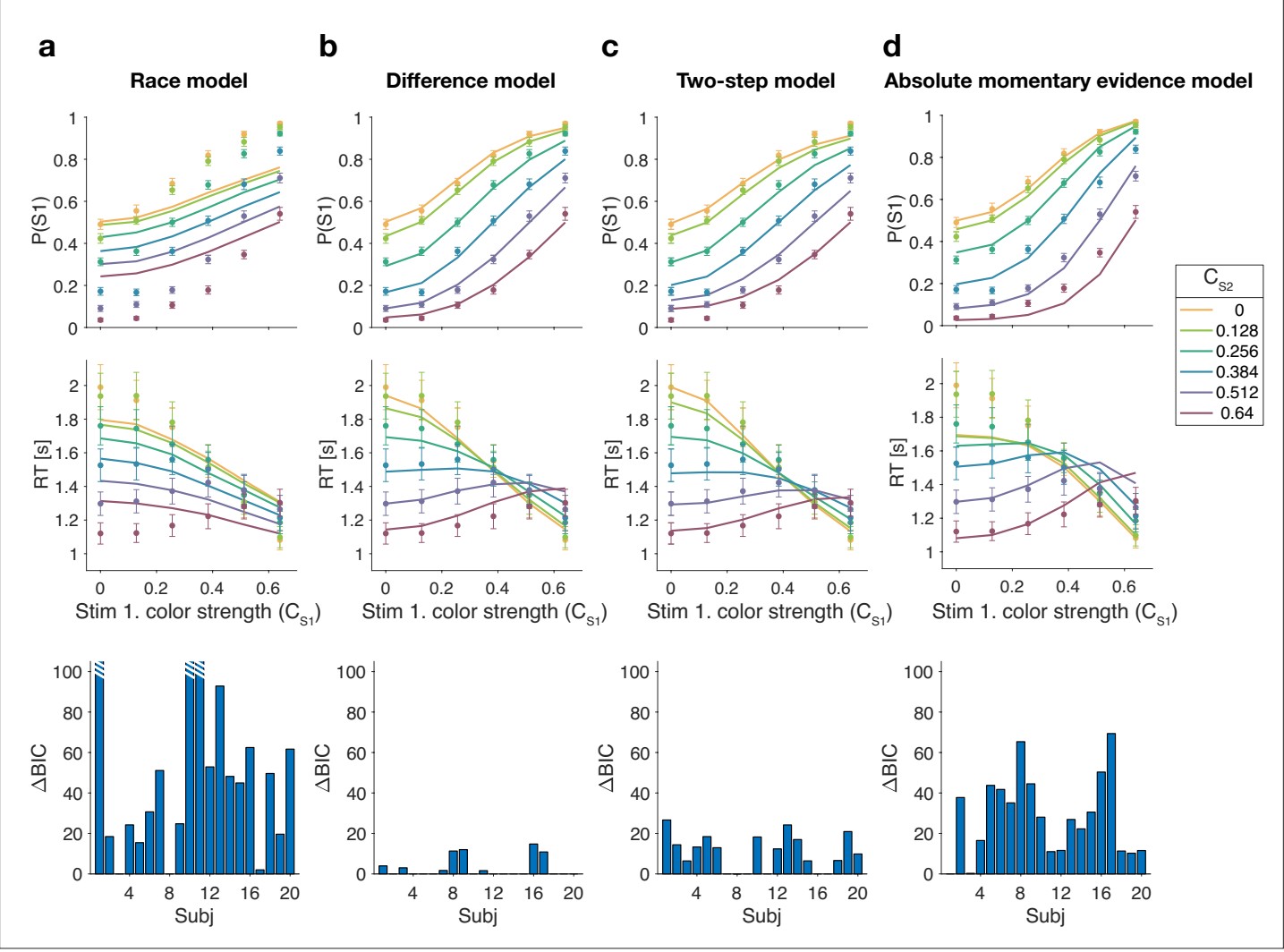

**Figure 3.** Reaction time difficulty judgment results (Experiment 1). Fits and model comparisons for (**a**) The race model, (**b**) the difference model, (**c**) the two-step model, and (**d**) the absolute momentary evidence model. Difficulty choices (top row) are shown as the proportion of trials in which participants chose S1 (left stimulus) as the easier stimulus. Reaction times are shown in the middle row. Data points are averages across participants (mean ± 1 SEM; $N = 20$). Lines represent model fits. Bottom row: $\Delta$BIC values for each model, relative to BIC value of the winning model for each participant (hatched bars represent values > 100).

*Figure 3* shows model fits, averaged across participants (for parameters, see *Supplementary files 2–5*). To compare the models we computed the Bayesian information criterion (BIC) for each model. This showed that the difference model provided the best fit overall (group-level $\Delta$BIC = 148.7 compared to the two-step model, which was the second best). Compared to all other models, the difference model was the preferred model for 12 out of 20 participants (*Figure 3*, bottom row). We also calculated the exceedance probability (*Stephan et al., 2009*; *Rigoux et al., 2014*), which measures how likely it is that any given model is more frequent than all other models in the comparison set, which across the four models gave [0, 0.97, 0.03, 0], showing that the difference model has a probability of 0.97 of being the most frequent model, compared to 0.03 for the two-step model.

The race model fails to capture the crossing of RTs for easy-easy vs. easy-hard stimuli since it predicts that decisions are fastest when both stimuli are easy. This model also fails systematically to explain the choice behavior (*Figure 3a*, top). Overall, the race model provides the poorest fit to participants' data (group-level $\Delta$BIC = 930.2 from difference model).

The absolute momentary evidence model provides a better fit to participants' choices and correctly predicts the crossing of RTs. However, it underestimates RTs when both stimuli are hard. This

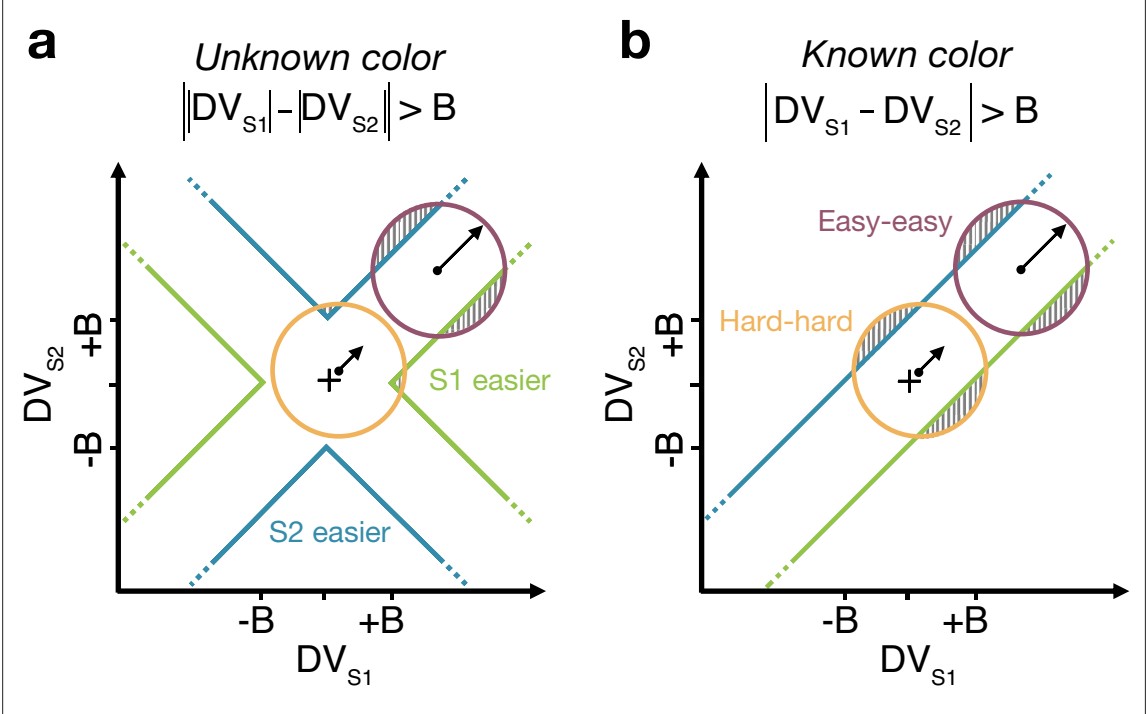

**Figure 4.** Reaction time task for unknown vs. known color dominance. (**a** and **b**) Schematic illustration of the unknown vs. known color tasks. For each condition, the dispersion of decision variables (DVs) is represented in the 2D space of $DV_{S1}$ (abscissa) and $DV_{S2}$ (ordinate), assuming a constant noise in the drift diffusion process independent of strength. To provide intuition colored circles show the dispersion expected without absorption at the bounds and represent contours of equal density of the decision variable. Green and blue lines represent decision boundaries (*B*) for S1 and S2 being easier. (**a**) In the unknown color condition, the difference model computes the difference between the two absolute DVs. (**b**) In the known color condition (example shown is for both stimuli blue dominant), the model compares the signed DVs, resulting in a larger region where the DV dispersion crosses the bounds (shaded areas). Thus, when both stimuli are blue-dominant, yellow (negative) evidence for S1 contributes to the decision that S2 is the easier stimulus.

is because in this model, any momentary evidence (regardless of sign) contributes to the difficulty decision. Consequently, this model also did not provide a good fit to the data overall (group-level $\Delta BIC = 509.0$ from best model).

Finally, the fits obtained with the difference model and two-step model were very similar, and both provided a good fit to participants' data. However, the two-step model requires an extra parameter for the initial color bound. Consequently, compared to the two-step model, the difference model was the preferred model for 14 out of 20 participants. Because of the similarity of these models, we verified that model recovery could distinguish between these models. We generated 200 synthetic data sets using each model and then fit them with both models. The fitting procedure correctly classified ($\Delta BIC > 10$) the correct model with 100% and 77% accuracy for the data generated by the difference and the two-step models, respectively (see Materials and methods).

In summary, we found support for a model in which difficulty is based on a moment-by-moment comparison of the absolute accumulated evidence from two stimuli.

## Difficulty judgments with unknown vs. known color dominance (Experiment 2)

The difference model predicts that behavior should change if the color dominance of each patch is known. *Figure 4a* contrast the difference model for an easy-easy vs. a hard-hard trial. When the drift component is small (hard-hard) the density of DVs across trials (orange circle) remains near the origin. When the drift component is large (easy-easy) the density moves into one of the channels (purple circle). Therefore, more density will cross the bound in the high drift case (gray hatched area for purple vs. orange circle), leading to RTs being shorter for the easy-easy condition compared to hard-hard condition (as in *Figure 3*). However, if the color dominance of both stimuli is known then

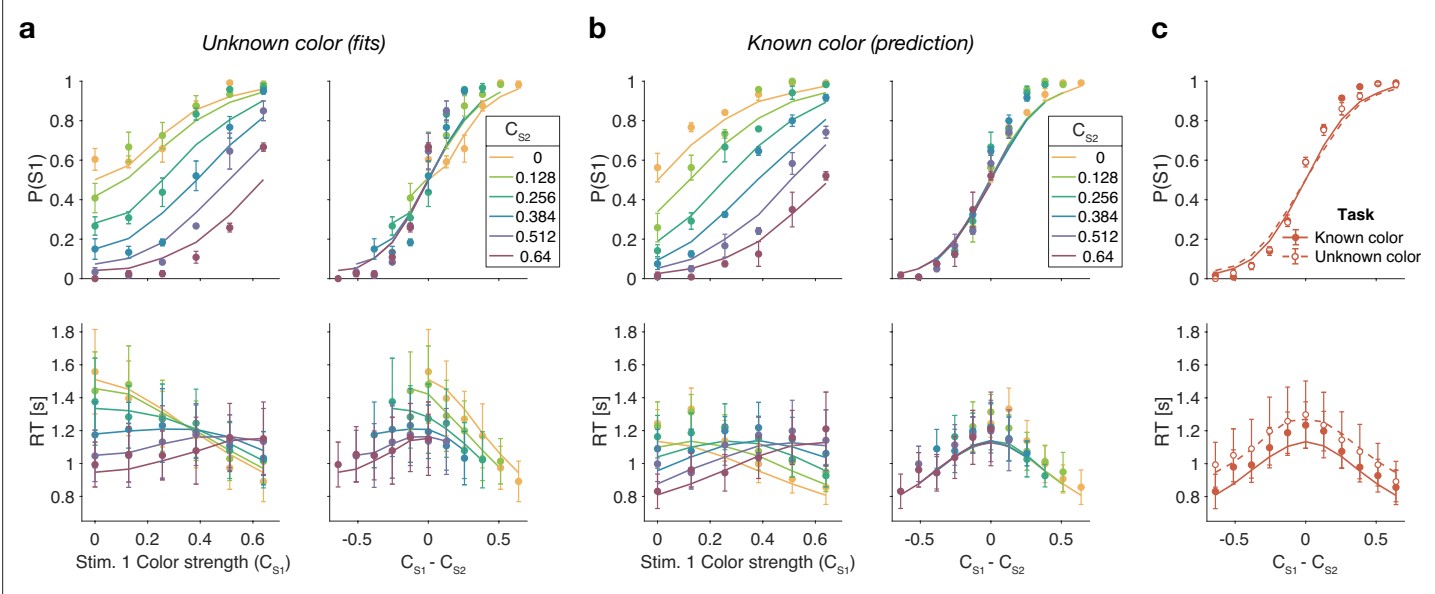

**Figure 5.** Reaction time task for unknown vs. known color dominance (Experiment 2a). (**a**) Unknown color condition. Proportion of S1 choices (top row) and reaction times (bottom row) plotted as a function of strength of S1 (left column) and difference of strengths of the two stimuli (right column). Lines show the fit of the difference model. (**b**) As (a) for the known color condition. Lines show the predictions of behavior when the color dominance is known (correct sign applied as in *Figure 4b*) based on the parameters obtained in the fit to (a). (**c**) Comparison of overall choice performance and reaction times (RTs) in the known vs. unknown color condition as a function of the absolute difference in strength levels (data mean ± 1 SEM across three participants).

this changes the bounds as shown in *Figure 4b*. In this example, both patches are known to be blue (positive) dominant. In this case the DVs can be compared in a signed manner where positive evidence corresponds to information in support of the known color, leading to a bound $DV_{S1} - DV_{S2} = ±B$, that is a simple channel (as in the two-step model). In this case both hard-hard and easy-easy trials will on average cross the bound at the same rate and faster in general than in the unknown color condition. When the dominant colors are known, (i) reaction times should be faster overall, and (ii) reaction times should only depend on the difference in strengths between the stimuli, that is on $\Delta C = |C_{S1} - C_{S2}|$ (i.e., reaction times for 0:0 and 0.64:0.64 strengths should be the same). Note that the two-step model becomes identical to the difference model if the color dominance of each patch is known, obviating the need for the first step of the model.

We test both of these predictions in an additional experiment on three participants, with the same basic paradigm as Experiment 1. However, in some blocks of trials, participants were informed that both stimuli would be blue dominant or both yellow dominant ('known color' condition). In the other blocks they did not receive any instructions about the stimulus color so that, as in Experiment 1, each stimulus could either be blue or yellow dominant ('unknown color' condition). For the unknown color condition, only trials in which both patches had the same color dominance were included in the analyses. This ensured better comparability with performance in the known-color blocks (where both stimuli by definition always had the same color).

We first replicate the findings from Experiment 1, as shown in the left column of *Figure 5a*. To evaluate the predictions, we also plot the data as a function of the difference in strength (*Figure 5a*, right column). This way of plotting the RT highlights the fact that the RTs depend on more than the difference in strengths (i.e., also the strengths of S2, colors). The curves are fits of the difference model.

*Figure 5b* shows the results for the same participants when the color dominance was known. The choice behavior (top row) is only subtly different from the unknown condition (*Figure 5a*) for reasons explained below. However, there is a striking, qualitative difference in the pattern of RTs. Consistent with prediction, the RTs appear to depend only on the difference in strengths (*Figure 5b*, bottom right), and they are faster when the color dominance is known (*Figure 5c*). Note that all solid curves accompanying the *known color* data in panels *Figure 5b and c* are not fits but predictions of the best fitting model to the *unknown color* condition.

To quantify the extent to which the difference in strengths ($\Delta C$) explains RTs in the known vs. unknown conditions, we compared the variance explained by the six unique $\Delta C$ levels for the two conditions. The increase in variance explained under the known color dominance is highly significant for all participants ($F_{176,1290} = 1.31, 1.41, 1.39$ all $p < 10^{-6}$). Further, in the known color condition only one of the three participants showed a significant explanatory effect of $C_{S2}$ on RT beyond its contribution to $\Delta C$ (p=0.002, 0.15, and 0.68 for the three participants; ANOVA of RT as additive in $\Delta C$ and $C_{S2}$, six levels each).

Examining choice as a function of the difference in strengths (*Figure 5c*) shows that accuracy of difficulty choices was similar in the two conditions (mean $P(\text{Correct}) = 0.87$, sd = 0.014 for known and 0.86, sd = 0.018 for unknown condition; Fisher's exact test, p>0.12 for each participant). However, RTs were significantly faster in known vs. unknown condition (mean = 1.08 s, sd = 0.19 vs. 1.15 s, sd = 0.31). A two-way ANOVA of RT by condition (two levels: known vs. unknown) $\times$ (six levels) gave a main effect of condition $p < 10^{-8}$ for all participants. Taken together, this suggests that similar bounds are used in the unknown and known color conditions as accuracy is similar. However, in the unknown color condition it takes longer, on average, to reach the bound (as in *Figure 4a* vs. *Figure 4b*) leading to longer RTs.

## Difference in confidence model

Up to now we have considered models that base difficulty on the color evidence, without requiring a decision about color dominance. We next consider the possibility that the difficulty comparison is actually a confidence judgment in disguise—that is the confidence one would assign to the two color dominance decisions were they made at each moment in time. Confidence is a mapping from DV and time into the log-odds of being correct if one were to choose the color (*Kiani and Shadlen, 2009*; *Kiani et al., 2014*). Our known color condition makes this seem unlikely as participants have no uncertainty about the color of each patch and, therefore, should have full confidence in both color estimates (so that the difference in confidence should be zero). It is not inconceivable, however, that participants evaluate a *counterfactual* form of confidence even in the known color condition—something to the effect of 'How confident would I be in the color choice if I did not know the color?'

We fit difficulty judgments to the unknown color condition by calculating the confidence for each decision, that is the probability (log-odds) of being correct if one were to choose the color dominance based on the sign of the DV. In the confidence model, the difficulty decision is made when the difference in absolute confidence for each stimulus reaches a bound. The fits of the confidence model are very poor ($\Delta\text{BIC} = 38, 56, 47$ for participants 1–3, in favor of the difference model; fit parameters:

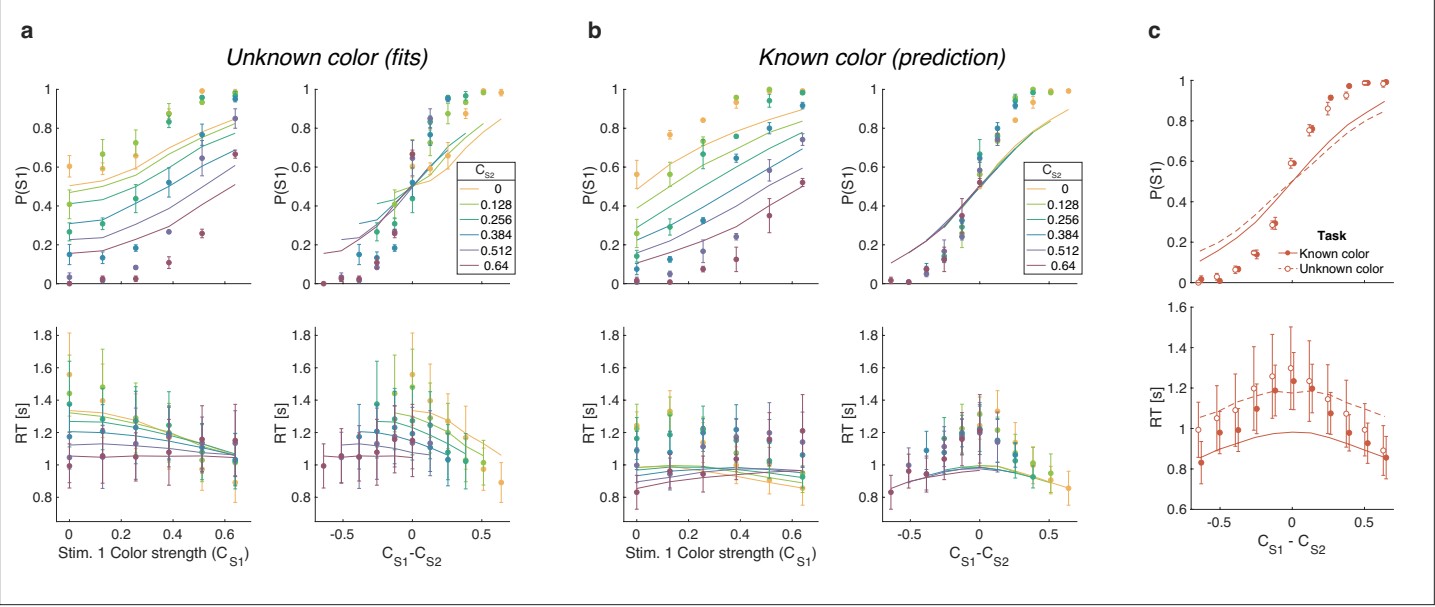

**Figure 6.** Same as *Figure 5* but with lines showing the fit of the confidence model to the unknown color condition and predictions for the known color condition.

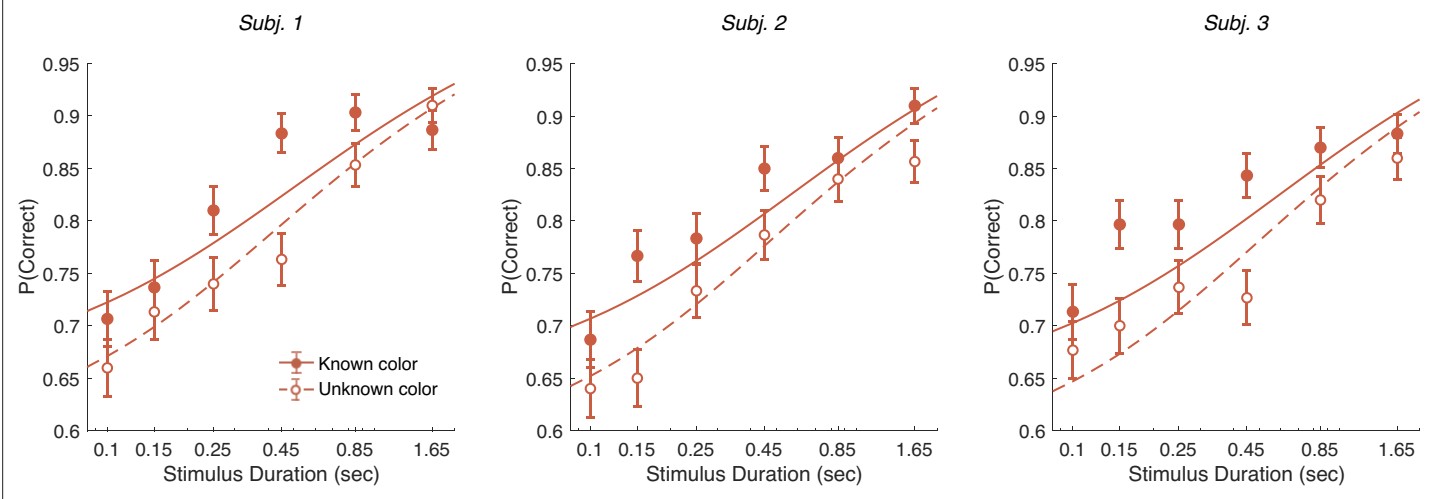

**Figure 7.** Controlled duration task for unknown vs. known color dominance. Performance in the controlled duration task for difficulty judgments in the known (solid-red) and unknown color condition (open-red). Participants' accuracy (mean ± 1 SEM) is shown for each stimulus duration. Here, we exclude trials which had no objective correct answer (i.e., trials in the difficulty task where S1 and S2 had the same strength). Lines illustrate model fits for the six stimulus times used in the experiment. Results are shown for individual participants.

*Supplementary file 8*), and the model fails to explain the known color condition (*Figure 6*). The reason for this failure may be understood as follows. In the difference model the decision only depends on the difference in the two DVs and not on the level of either DV. In contrast, the difference in confidence depends on the difference in DVs as well as the actual level of each DV. Indeed, confidence increases supra-linearly with DV such that a 0.64:0.64 coherence trial will tend to have a bigger difference in confidence than a 0:0 trial, which would lead to much shorter RTs for the former. This leads to the confidence model failing to explain the range of RTs apparent in the data (*Figure 6*).

## Controlled duration task

In addition to the RT version of the task, the participants performed a version of the task in which viewing duration was controlled by the experimenter. Stimulus duration was sampled from a truncated exponential distribution leading to an equal probability of six discrete durations: $\{0.1, 0.15, 0.25, 0.45, 0.85, 1.65\}$ s (which leads to roughly equal changes in accuracy between consecutive duration). We focus on short duration stimuli as the model predicts that differences in performance between the known and unknown color dominance conditions are largest for short durations. Participants had to wait for the stimuli to disappear before indicating their response. Again there were blocks with unknown color dominance and blocks with known color dominance. The difference model predicts that accuracy should be better in the known color dominance condition.

*Figure 7* shows the decision accuracy as a function of stimulus duration for the difficulty tasks, for both known (solid red) and unknown (hollow red) color. As expected, accuracy improved with longer stimulus durations. More importantly, choice accuracy was generally higher in blocks with known color (accuracy across durations: mean $P(\text{Correct}) = 0.82$, sd $= 0.01$) compared to blocks with unknown color (mean $P(\text{Correct}) = 0.76$, sd $= 0.01$; Fisher's exact test, $p < 0.001$ for each participants).

We fit a single DDM model simultaneously to each participant's choices for both difficulty tasks (solid and dashed lines in *Figure 7*). For the unknown color task we used the difference model as in Experiment 1 (i.e., using the equation shown in *Figure 4a*). For the known color task, we set the signs of the DVs according to the true color dominance (as in *Figure 4b*).

The model has four parameters (fit parameters *Supplementary file 7*): a drift rate coefficient ($\kappa$) and three parameters controlling how each decision bound collapses over time (as in the RT experiments). Previous work has shown that although evidence integration is serial, there is a buffer that can store a limited amount sensory information from both streams of evidence (*Pashler, 1994*). This means information can be acquired in parallel until the buffer is full and then the accumulation happens in a multiplexed manner (*Kang et al., 2021*). We included a 80 ms buffer in the model, that represents the duration of the two stimulus streams that can be held in short-term memory. The buffer simply

accommodates the observation that decision makers use all the information from both stimuli when they are presented for very short durations. Indeed, we found that the model with buffer was superior to a model without a buffer for all three participants ($\log_{10} \text{BF} = 2.0, 2.9, 5.9$ in favor of the model that includes the buffer).

An alternative explanation for the accuracy improvements is that the difficulty judgment is always based on the absolute DVs (as in *Figure 4a*) but that the drift rate might be higher when participants know the correct color. For example, integration might be more efficient when focusing on evidence that is consistent with the instructed color. We therefore compared our model with an alternative model in which we allowed two different $\kappa$ parameters—one for the known color and one for the unknown color condition. Model comparison reveals that the more parsimonious model with a single $\kappa$ across both conditions was preferred overall and strongly for two of the participants: $\log_{10} \text{BF} = 8.5$ (group level) and $3.7, 4.1, 0.75$ (participants) in favor of the single $\kappa$ model.

In summary, Experiment 2 provides evidence that knowing the correct color improves the accuracy and speed of difficulty judgments despite the fact that color identity is not relevant for the final difficulty choice. This effect is explained by the difference model: when the correct color is known, difficulty judgments are based on a comparison of appropriately signed DVs, which provide more information than the unsigned absolute strength of evidence, as explained above.

## Optimal model

We model the decision process on of Experiment 1 (RT) as a partially observable Markovian decision process (POMDP), and transform it into a fully observable Markov decision process (MDP) over the decision maker's belief states. The belief states are uniquely defined by the tuple $\langle t_{S1}, t_{S2}, DV_{S1}, DV_{S2} \rangle$ where $t_x$ is the time elapsed sampling stimulus $x$ and $DV_x$ is the accumulated evidence for stimulus $x$. Three actions are available in each belief state: choose the option $S1$, option $S2$, or continue gathering

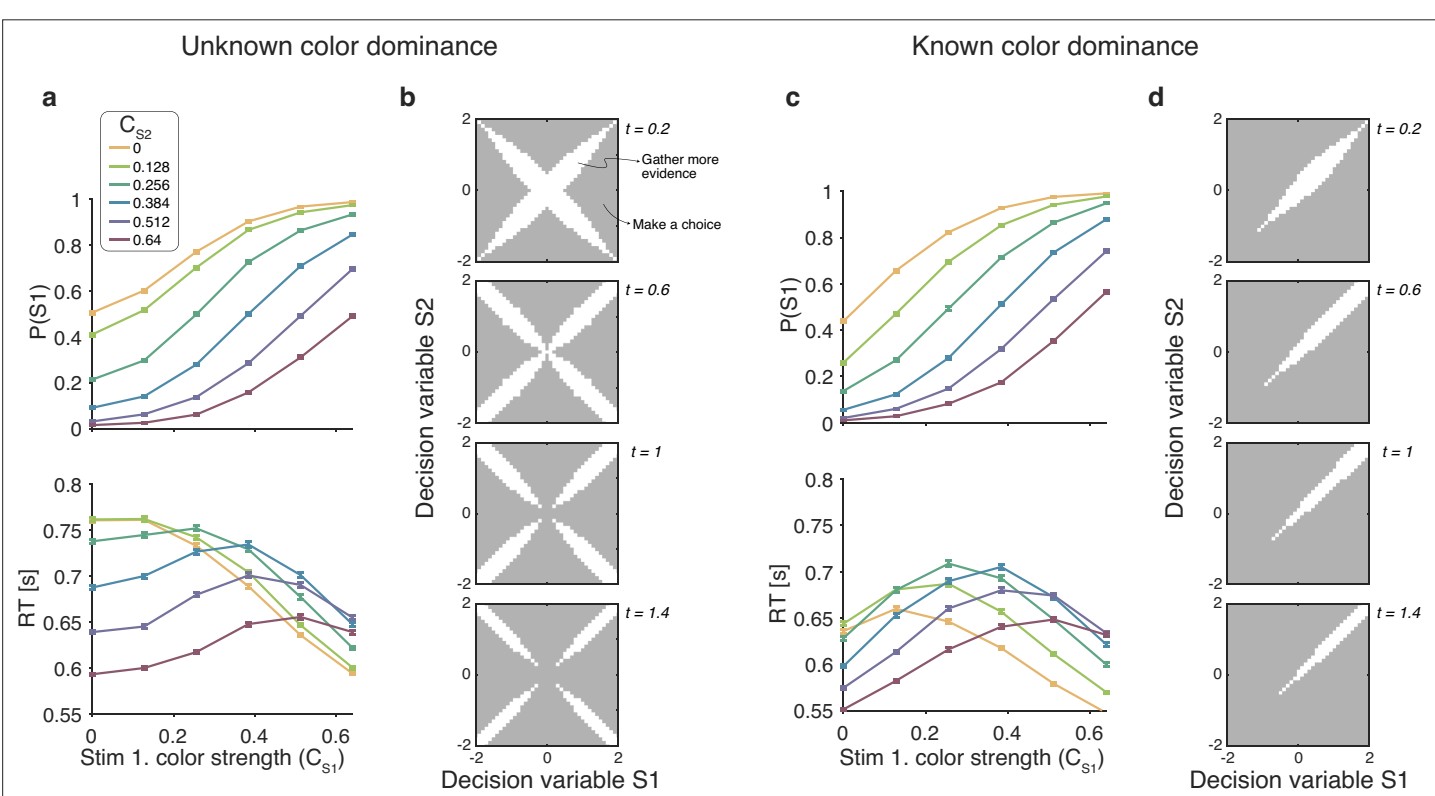

**Figure 8.** Optimal policy for difficulty decisions. (**a**) Choices and response times obtained from simulations of the model that maximizes the rate of correct choices, derived for the reaction time version of the task with unknown color dominance (as in Experiment 1; N=200,000 simulated trials). (**b**) The optimal decision policy is a deterministic mapping from a belief state to an action. The figure identifies the values of the decision variables for stimuli S1 and S2, for which it is optimal to continue sampling sensory information (shown in white) or commit to a choice (shown in gray). Each panel represents different time within a trial (where the time spent sampling each stimulus is *t/2*). (**c** and **d**) Same as panels (**a** and **b**), but for the version of the reaction time task in which the color dominance is known (as in Experiment 2).

evidence (with evidence time shared equally between $S1$ and $S2$). The assignment of policies to belief states is deterministic: only one action is chosen in each state. Transitions between states, however, are stochastic, and depend on the coherences of the stimuli and noise, which is assumed Gaussian.

For the unknown color dominance condition (Experiment 1) behavior derived from the optimal decision policy (see Materials and methods) is qualitatively similar to that of the participants (*Figure 8a*). Optimal performance shows a clear modulation by the strength of each stimulus. The proportion of correct responses and the decision speed are higher when strength is higher. The optimal model also captures the crossing of RTs for easy-easy vs. easy-hard stimuli that we observed in the data.

To determine which of the four models presented above is most similar to the optimal model, we performed simulations of the optimal model and fit the four models to the simulated data. The model that best accounted for the simulated data was the difference model, followed by the absolute momentary evidence model ($\Delta BIC = 2,847$ relative to the difference model), and the two-step and race models ($\Delta BIC = 6,949$ and $15,465$ respectively).

An analysis of the decision space of the optimal model allows us to identify similarities with the four models. In *Figure 8B* we show the decision space of the optimal model for four different times ($t_{S1} + t_{S2}$). At early times, the decision space resembles that of the difference model, in that the decision variable can diffuse along four paths defined by the possible signs of the color coherences. However, later in a trial, the regions where it is optimal to continue sampling evidence become disjoint. That is, the center of the graph is no longer a region where it is optimal to keep sampling information. This is because, if after prolonged deliberation the decision variable is still in the central region of the decision space, then it is reasonable to infer that the sensory evidence is weak, in which case it may be optimal to hasten the decision and move to the next trial rather than continue deliberating on a decision that has a high probability of being wrong. The logic is the same as why it is optimal to collapse decision boundaries over time in binary decisions (*Frazier and Yu, 2007*; *Drugowitsch et al., 2012*). Interestingly, such a disjoint decision space approximates an internal commitment to a decision about the sign of the color coherence, as in the two-step model.

We also derived the optimal model for the known color RT condition of Experiment 2. To do this we limited the distribution of coherences so that both were positive. The optimal model was in qualitative agreement with the experimental data (*Figure 8c*) and its bounds (*Figure 8d*) are similar to the difference model. Response times were largely determined by the difference between the coherences, such that RTs were faster when the absolute value of difference was higher. However, there is a notable exception. In contrast to what we observed in the experimental data, the RTs of the optimal model also depended on the absolute value of the coherences, such that coherences closer to the extremes of the range led to faster RTs. This can be understood as follows. Near the range limit of coherences, the decision maker can obtain evidence samples that are very informative about the coherence of a patch. For example, if a very negative sample is obtained for one of the stimuli, then, since the decisions maker knows that all coherences are positive, then this evidence value is most likely due to a very low coherence, from which one could conclude that the other stimulus is likely to be of higher coherence. In contrast, for coherences that are in the middle of the range, there is no single sample that can be as informative. The optimal model—but not the participants—seems to exploit the knowledge of the distribution over coherences to make better decisions.

## Discussion

Tasks that require consideration of multiple samples of evidence serve to elucidate the cognitive and neurophysiological mechanisms of decision making (*Gold and Shadlen, 2007*; *Brody and Hanks, 2016*). They promote the framework of sequential sampling with optional stopping, which unifies accounts of choice accuracy, response time, and confidence (*Kiani et al., 2014*; *van den Berg et al., 2016*). In this paper we build on this sequential sampling framework to understand how people construct a subjective estimate of the difficulty of a decision. In many circumstances, one might judge the relative difficulty of two tasks by performing them and comparing accuracy, confidence, and decision time—in a word, experience. We were intrigued by the possibility that relative difficulty of two decisions could be determined directly by accumulating a transformed version of the evidence used to make the perceptual decisions. We show that this is possible and, moreover, the process is distinct from the perceptual decisions. In some cases, the difficulty decision is made faster than at least one of the perceptual decisions (e.g., the difficult perceptual decision when the other stimulus is very

easy). This is true despite the difficulty decision requiring monitoring of two patches (which might be expected to be slower than monitoring one patch). For example, making a color choice for 0% color strength takes longer than a difficulty choice for 0:0.52 color strengths. Thus, the difficulty judgment does not require completion of the color decisions. However, in other cases, the two perceptual decisions are almost certainly completed before the difficulty decision (e.g., when color decisions are both very easy).

Studies of subjective beliefs, such as difficulty and confidence, often rely on numerical scales or discrete categorizations (e.g., *Yildirim et al., 2019*; *Ais et al., 2016*; *van den Berg et al., 2016*). This is in some ways more intuitive and more germane, perhaps, to real-world judgments, which do not always warrant a comparison (but see *Gweon et al., 2017*). We pursued the comparison as it allowed us to relate the difficulty decision to quantities that are known to govern the simple decision. Moreover, comparative judgments of difficulty allow us to ignore participant-specific attributes of the estimation (e.g., a task may be more difficult for one participant than for another one), and the idiosyncrasies involved in mapping a subjective quantity to a numeric scale (*Mamassian, 2020*). Measuring the time it took our participants to form their judgments of difficulty allowed us to test between different mechanisms.

The model that best explained the difficulty choices and RTs (difference model) was one in which participants accumulate information about the predominant color—which would be used to resolve each low-level decision—and the decision about difficulty is based on the comparison of the accumulated evidence for the two color decisions. The decision about difficulty terminates when the difference between the absolute values of the evidence accumulated by the low-level decisions crosses a threshold. The model makes concrete the idea of difficulty estimation as a metacognitive process, as the evidence for the difficulty judgment is given by the output of lower-level decisions, thus instantiating a processing hierarchy.

The difficulty decision relies on a difference in the absolute values of the DVs that would be formed to make the individual decisions. This quantity might be computed using a simple *max* operation.

Neural implementation of simple drift diffusion, associated with a single binary decision, is organized as a race between two drift diffusion processes: (i) the accumulation of the difference in momentary evidence for blue minus yellow (or more generally, colors that are not blue) and (ii) yellow minus blue (or not yellow). These racing accumulations are anticorrelated, albeit imperfectly. The absolute value of the DV can be approximated by the greater of the competing accumulations. The same computation is also applied to the other stimulus and the difference between the absolute DVs for the two stimuli is then computed. Such a process could be extended to difficulty decisions over more than two stimuli.

## Difficulty judgments and the magnitude effect

The RT for difficulty judgments did not just depend on the difference in difficulty between the two color decisions. Specifically, responses were faster, when the sum of the color strengths was larger (i.e., a 'magnitude' effect) even if the difference in difficulty was the same. This effect has a parallel in value-based decision making. For instance, when people choose between two highly desirable items, the decision is faster than when the items are both less desirable, even if the decisions are difficult because the pairs comprise items of approximately equal value (*Smith and Krajbich, 2019*). This effect can be explained by arguing that attention fluctuates between both items under comparison and that attention has a multiplicative effect on the subjective value of the unattended item (*Smith and Krajbich, 2019*; *Sepulveda et al., 2020*). Because of this multiplicative effect, the greater the value of the items under comparison, the greater the discount of the unattended item, leading to faster decisions (i.e., the magnitude effect).

This explanation does not apply in our case, since we could largely abolish the magnitude effect by informing the participants about the dominant color in each stimulus patch. In our task, the magnitude effect occurs, because DVs on hard-hard trials (*Figure 4a*, orange circle) cross the bounds later than on easy-easy trials (purple circle). This arises because the DVs for easy-easy trials move into one of the channels, whereas weak-weak DVs linger near the origin making them less likely to cross a bound. When the color dominance is known (*Figure 4b*), the bounds change so that the DV can reach a bound when lingering near the origin. Therefore, the RTs now only depend on difference in difficulty and not the magnitude.

## Optimal model of difficulty judgments

The model that maximizes reward rate in our unknown color dominance task (Experiment 1) qualitatively reproduces the behavior of the participants. In particular, the optimal model shows a similar 'magnitude effect' to that observed in the data and the same criss-cross pattern in response times. Therefore, these features of the data do not signify suboptimal decision making. We fit the four models (*Figure 2*) to simulations of the optimal model and found that the difference model best accounts for the data from the optimal model. That is, the model that most closely resembles the optimal model is also the one that best explains the participants' data. The decision space derived from the optimal policy was similar to that of the difference model, with one notable exception. The optimal model predicts that the speed at which the bounds collapse depend both on elapsed time and the magnitude of the decision variables (*Figure 8b*). The same was observed in the optimal model (*Figure 8d*) derived for the known color dominance task (Experiment 2).

In contrast, in the difference model the speed at which the bounds collapse depends only on elapsed time. It remains to be determined the extent to which this difference has a meaningful impact on behavior. Conducting a variant of our experiment in which participants are incentivized to maximize reward rate could be informative.

## Parallel vs. serial processing of the two stimuli

In our models we assume that participants sample the two stimuli in difficulty judgment sequentially through alternation. However, our results are not affected by whether participants sample the stimulus sequentially through alternation (which we assume is fast and has equal times for both stimulus) or in a parallel manner (cf., *Kang et al., 2021*). What does change is the parameters of the model (but not their predictions/fits). In the parallel model information is acquired at twice the rate of the serial model. We can, therefore, obtain the parameters of parallel models (that had identical predictions to the serial model) directly from the parameters of the current sequential models simply by adjusting the parameters that depend on the time scale (subscripts $s$ and $p$ for serial and parallel models): $\kappa_p = \kappa_s/\sqrt{2}$, $u_p = u_s\sqrt{2}$, $a_p = a_s/2$, and $d_p = 2d_s$ (*Equation 2*).

## Potential limitations

We acknowledge several limitation to our study. First, we only consider accumulation models as these have been very successful in explaining choice, RT, and even confidence in many perceptual and mnemonic decision tasks, including color discrimination (*Bakkour et al., 2019*; *Gold and Shadlen, 2007*). Using such a model we were able to account for difficulty judgment in both RT and controlled duration tasks in which the color dominance was either unknown or known. Given that simple perceptual decision of the type we study have been shown to involve accumulation, we chose not to consider non-accumulation models (*Stine et al., 2020*; *Cisek et al., 2009*). Second, for Experiment 2 we required considerable data from each participant to be able to test our hypothesis. Therefore we ran 3 participants over 18 sessions each. While this provided a large data set we accept that the number of participants is small. Third, it is an open question whether the results we obtain from our simple perceptual decision task would generalize to more naturalistic real-world tasks (e.g., choosing an instrument to learn or a recipe to cook). We chose to study judgments of difficulty in simple tasks amenable to quantitative modeling and, eventually, neurophysiological investigations in nonhuman animals. Moreover, the feedback we provided on each trial—which of the two tasks was actually easier—is likely to differ in the real world, where difficulty is rarely directly indicated. Importantly, accumulation models have been extended to more complex tasks such as decision making in a game of chess (*Fernandez Slezak et al., 2018*) and therefore in principle the same types of algorithm could operate for more complex difficulty judgment tasks.

## Confidence does not underlie difficulty judgments

It has been proposed that confidence judgments about the accuracy of a decision require having learned a mapping between (i) the state of a decision variable ($DV$) and elapsed time ($t$) and (ii) the probability that the decision is correct (*Kiani and Shadlen, 2009*; *Kiani et al., 2014*; *van den Berg et al., 2016*). This mapping is used to define a policy; for instance, respond with high confidence if the estimated probability of being correct is greater than a criterion. An explicit calculation of confidence in the color decisions is not necessary in our task, because the difficulty decision is based on quantity

derived directly from the two color decision variables. Indeed a model that compares confidence to make the difficulty judgment provides a poor fit to the data (*Figure 6*). That said, there may be situations where the difficulty determinations involve calculation of confidence as an intermediate step. For example, in a difficulty comparison of a color dominance in one patch and motion direction in another patch, the decision variables may not be directly comparable and may therefore require a conversion to confidence (*de Gardelle et al., 2016*).

In summary, we have shown that a difference model can explain difficulty judgments in both an RT and controlled duration task and when the dominant color of both patches is either unknown or known. The results extend decision-making models, which have been used to explain choice, RT, and confidence to judgments of difficulty.

## Materials and methods
### Participants
For Experiment 1 (Difficulty judgments in an RT task), 51 participants were recruited on Amazon Mechanical Turk. Only participants who completed the entire study and met performance-based inclusion criteria (see below) were included in the final sample of 20 participants (12 male and 8 female; 19 right-handed; age 20–51, mean = 33.6, sd = 9.1). Participants completed two 1 hr sessions each. They received $1 for each session, plus a performance-based bonus of up to $5. Participants who successfully completed both sessions of the task within 48 hr received an additional bonus of $1. These experiments were an early foray into using Mechanical Turk and we simply matched payment with the typical payment for similar online experiments. We have since become aware, and agree with, advocates who feel the pay is too low and have since moved to using the Prolific platform and ensure we pay $8/hr minimum.

For Experiment 2 (Difficulty judgments with unknown vs. known color dominance), four participants were recruited via a Sona Systems participant pool. After initial training, three participants (one male and two female; all right-handed; age 20–24, mean = 21.7, sd = 2.1) were selected for the main experiment based on their performance (see below). Participants completed a total of eighteen 1 hr sessions and received $17/hr and an additional performance-based bonus.

All participants had normal or corrected-to-normal vision and were naïve about the hypotheses of the experiment. Participants provided written informed consent prior to the study. The study was approved by the local ethics committee (IRB-AAAR9148, Institutional Review Board of Columbia University Medical Center).

### Apparatus and stimuli
Both experiments were conducted remotely during the SARS-CoV-2 pandemic (summer 2021). Participants completed the task online using a Google Chrome browser. The task was programmed in JavaScript and jsPsych (*de Leeuw, 2015*). During the task, two dynamic random dot patches with yellow and blue dots were presented in rectangular apertures ($3 \times 5°$, horizontal × vertical) to the left and right of a red fixation cross, separated by a central gray bar ($2 \times 5°$; *Figure 1a*). Visual stimuli were presented with a screen refresh rate of 60 Hz and stimulus density of 16 dots/deg$^2$/s (i.e., 4 dots displayed in each aperture on each frame). On each video frame, each dot was displayed in a location chosen from a uniform distribution over the aperture. The color of each dot was determined independent of its location, according to one of six color strength levels (see below).

Prior to the experiment, participants completed a virtual chin-rest procedure in order to estimate viewing distance and calibrate the screen pixels per degree (*Li et al., 2020*). This procedure involves first adjusting objects of known size displayed on the screen to match their physical size and then measuring the horizontal distance from fixation to the blind spot on the screen (taken as $13.5°$).

### Overview of experimental tasks
On each trial, two patches of random dots were presented after an onset delay of 400–800 ms. Participants were asked to decide for which one of two patches of dots it is easier to decide what the dominant color is (yellow/blue). The instructions given the participants was 'Your task is to judge which patch has a stronger majority of yellow or blue dots. In other words: For which patch do you find it easier to decide what the dominant color is? It does not matter what the dominant color of the

easier patch is (i.e., whether it is yellow or blue). All that matters is whether the left or right patch is easier to decide.'

Participants were instructed to press the F or J key with their left/right index finger to indicate whether the left or right stimulus was the easier one, respectively. Participants did not have to report the individual color decision (yellow or blue) for either stimulus. Instead, they were required to indicate which patch (left or right) was the easier one, regardless of whether the majority of dots in that patch was yellow or blue.

The difficulty of the color choice was conferred by the probability that a dot would be colored blue or yellow on each frame. We refer to the signed quantity, $C^{\pm} = 2(p_{blue} - 0.5)$, as the color coherence where positive coherences refer to blue dominant stimuli. The dominant color of each stimulus (yellow/blue) and its difficulty (color strength $C = |C^{\pm}|$) was fixed during a trial but randomized independently across trials. We used six different coherence levels $\pm\{0, 0.128, 0.256, 0.384, 0.512, 0.64\}$, resulting in 12 signed coherence levels for the two colors. All 12 × 12 color-coherence combinations for the two patches were presented in a pseudo-random, counterbalanced manner during the task.

Visual feedback was provided at the end of each trial. For correct responses, participants won 1 point. After errors and miss trials (too early/late), participants lost 1 point. For trials in which both stimuli were equally difficult (i.e., same coherence level), half of the trials were randomly designated 'correct'. Miss trials were repeated later during the same block. Participants were instructed to try and gain as many points as possible and at the end of the experiment they received an extra bonus of 1 cent for every point they accumulated. Their point score was shown in the corner of the screen throughout the task and additional feedback about percent accuracy was provided at the end of every block.

Prior to completing the task with difficulty judgments, all participants were first trained on a task in which they had to make color judgments (blue/yellow) about a single patch of dots presented to the left or right of the central fixation cross. Participants used the M and K keys with their right index/ middle finger to indicate their response. Throughout the task, the response mapping (M=yellow, K=blue) was shown on the screen. The ± sign of the 0 coherence level determined which response would be rewarded.

Participants were instructed to keep their eyes fixated on the central fixation cross throughout the task.

## Experiment 1: Difficulty judgments in an RT task

Participants performed two separate sessions with a total of 432 trials of the color judgment task and 1152 trials of the difficulty task. In session 1, participants were first trained on the color judgment task (see above). They performed 6 blocks of 72 trials each, in which all 12 signed coherence levels were presented in random order. They then completed 3 blocks of the difficulty judgment task (96 trials/ block), in which all 12 × 12 coherence combinations for the two patches were presented in random order. In a second session there were 9 blocks (96 trials/block) of the difficulty task.

In both sessions, participants performed an RT task in which they were instructed to respond as soon as they had made their decision. The stimuli disappeared as soon as participants made a response. Participants were instructed to try to be both as fast and as accurate as possible in order to maximize their score. Warning messages were presented if participants initiated a response before stimulus onset or within 200 ms of stimulus onset ('too early') or when RTs exceeded 5 s ('too slow!').

### Exclusion criteria

Online experiments pose challenges concerning quality control (*Chmielewski and Kucker, 2020*). Therefore, we excluded participants who could not perform the task sufficiently well. After session 1, we fit a logistic of color choices against coherence and difficulty choices against the difference in strength of S1 and S2. In order to ensure high data quality, participants with low choice accuracy were excluded from the experiment and were not invited to participate in session 2 of the experiment. Specifically, participants whose sensitivity (slope parameter of logistic) of color choices was less than 4 (n=4), or whose sensitivity of difficulty choices was less than 3 (n=21) (a lower threshold used here as we found difficulty choices were harder than color choices), were not invited to participate in session 2 and were excluded from all analyses. In addition, participants (n=6) were excluded from all analyses because they did not complete session 2 despite meeting the performance criteria. Importantly,

exclusion criteria were not based on RT (our main analysis in the study), and instead, were strictly based on accuracy to ensure that participants followed the task instructions correctly. In the final sample, mean sensitivity of color choices was 9.67 (sd = 2.66) and mean sensitivity of difficulty choices was 5.24 (sd = 1.03).

## Experiment 2: Difficulty judgments with unknown vs. known color dominance

Participants completed a total of 18 sessions on separate days. The first 4 sessions were regarded as training (see below) and were not included in the analysis. During sessions 5–17, participants performed different versions of the difficulty judgment task that alternated every 3–4 blocks of 96 trials each (order counterbalanced across participants): (i) controlled duration task with unknown color (total of 3888 trials per participant), (ii) controlled duration task with known color (1944 trials, half of which were blocks with blue stimuli while the other half of blocks had yellow stimuli), (iii) RT task with unknown color (2592 trials), (iv) RT task with known color (1296 trials, half of which were blocks with blue stimuli while the other half of blocks had yellow stimuli). In the final session, participants completed 600 trials of a controlled duration task in which they had to judge the dominant color of a single stimulus. In all versions of the controlled duration task, stimuli were presented with one of six stimulus durations $\{0.1, 0.15, 0.25, 0.45, 0.85, 1.65\}$ s, which were randomized within blocks. This gives a discrete sample from a truncate exponential distribution. Participants were instructed to wait for the stimuli to disappear before indicating their response. Warning messages were presented if participants initiated a response before stimulus offset ('too early') or later than 5 s after stimulus offset ('too slow!').

In the difficulty task with unknown color, all $12 \times 12$ signed coherence combinations for the two patches were presented in random order. Participants were instructed that each color patch could be either yellow or blue and that the two patches would not necessarily be of the same color. In the blocks with known color, only coherence combinations with the same sign ($6 \times 6$ combinations) were presented. Participants received instructions that all patches in the following blocks would be dominantly blue (or yellow). Throughout the task, participants were shown a brief instruction in the corner of the screen reminding them of the condition of the current block.

In all versions of the task, coherence combinations where both stimuli had the same difficulty (i.e., same strength) were presented with one third the frequency compared to other coherence combinations in order to reduce overall trial numbers. The rationale behind this was that choice accuracy was determined randomly in this condition, and thus, it is not informative with regard to participants' actual choice performance. Consequently, for analyses of choice accuracy in the controlled duration task, all trials with equal coherence combinations were excluded. All other coherence combinations were presented with equal frequency and in a randomized order.

Although all possible $12 \times 12$ coherence combinations were presented in the unknown color condition, only trials in which both patches had the same color were included in the analyses. This ensured better comparability with performance in the known color blocks (where both stimuli by definition always had the same color). The task was designed with this in mind due to the fact that previous pilot studies from our lab revealed that performance tends to be worse when the two patches are of different color compared to when they are of the same color. Thus, participants performed twice the number of total trials with the unknown color condition, but only trials with equal color combinations were included in the main analyses, resulting in the same trial numbers for the known vs. unknown condition (1944 trials each for controlled duration and 1296 trials each for RT).

### Training sessions

Prior to the experimental sessions, participants completed four training sessions. During session 1, participants were trained on a controlled duration task with single color judgments in which stimulus durations were drawn randomly from an exponential distribution with $\mu$ = 800 ms, truncated at 400–1800 ms. In session 2, participants performed an RT version of the color judgments task. Finally, in sessions 3 and 4, participants performed the difficulty judgments task with half the blocks of each session being an RT version of the task while the remaining blocks were a controlled duration version in which stimulus durations were drawn randomly from an exponential distribution (session 3: $\mu$ = 900 ms, truncated at 400–2000 ms; session 4: $\mu$ = 800 ms, truncated at 200–1800 ms). After training,

all four participants met the performance criteria (slope of logistic function > 4 for color choices and > 3 for difficulty choices). However, one participant was excluded from the remaining sessions of the experiment due to failing to complete the training sessions in a timely manner.

## Drift diffusion model of color judgments

We fit a standard drift diffusion model to participants' color choices and RTs. The model assumes that MCE is accumulated into a DV. The process terminates when the DV reaches an upper or lower bound ($\pm B$), which corresponds to making a blue vs. yellow choice, respectively. The DV is determined by a Wiener process with drift, which starts at 0 and then evolves according to the sum of a deterministic and a stochastic component (with discrete updates every $\Delta t$):

$$\Delta DV_t = \mu \Delta t + \mathcal{N}(0, \sqrt{\Delta t}) \tag{1}$$

The deterministic term depends on the drift $\mu = \kappa(C^{\pm} + C_0)$, where $C^{\pm}$ is the signed color coherence (positive for blue; negative for yellow), $\kappa$ converts coherence to drift rate, and $C_0$ allows for a color bias. Here, the bias term is modeled as an offset in the coherence, rather than a shift in the starting point of the accumulation process. This approximates the optimal way of implementing a bias when coherence levels vary across trials (*Hanks et al., 2011*; *Zylberberg et al., 2018*).

The second term of *Equation 1* describes the stochastic component, which captures the variability introduced by the noise in the stimulus and in the neural response. This variability is modeled as independent samples from a Normal distribution with mean 0 and standard deviation $\sqrt{\Delta t}$, which results in the variance of the DV being equal to 1 after accumulating evidence for 1 s. This was done by convention, since for any other scaling of the variance, it would be possible to define an equivalent model in which the variance is 1 and the other parameters are a scaled version of the original ones (*Palmer et al., 2005*).

The accumulation process terminates when the DV crosses one of two bounds ($\pm B$), resulting in a blue or yellow choice. The decision time $T_d$ is the time it takes for the DV to reach a bound. To account for the observation that RTs tend to be slower in erroneous, compared to correct choices, for a given coherence level (data not shown here), we implemented bounds that collapse over time (*Rapoport and Burkheimer, 1971*; *Drugowitsch et al., 2012*; *Shadlen and Kiani, 2013*). Collapsing bounds result in an increased probability that the *DV* will reach the wrong bound the longer the accumulation process takes. We parameterized the upper collapsing bound as a logistic function:

$$B(t) = \frac{u}{1 + e^{a(t-d)}} \tag{2}$$

with three parameters $a, u, d$. Therefore, the bound collapses (slope related to $a$) and reaches a value of $u/2$ at $t = d$ and approaches 0 as $t \to \infty$. The lower bound is simply the negative of the upper bound ($u \to -u$).

Given a set of parameters ($\Phi = [\kappa, C_0, u, a, d]$), we can estimate the joint probability density function for choices and decision times $T_d$ as a function of the signed color coherence $C\pm$. We used a $\Delta t$ of 0.5 ms and obtained the probability density function by numerically solving the Fokker-Planck equation associated with a Wiener process with drift (*Kiani and Shadlen, 2009*), using the finite difference method of *Chang and Cooper, 1970*. Finally, the RT is determined by the sum of the decision time $T_d$ and a non-decision time, which is assumed to be Gaussian with mean $T_{nd}$ and a standard deviation that was fixed to $\sigma_{Tnd} = 0.05$ s.

We fit the model parameters to the mean choice-RT data of individual participants by maximizing the likelihood of observing the data given the model parameters ($\Phi = [\kappa, C_0, u, a, d, T_{nd}]$) and the signed color coherence $C\pm$. We used Bayesian adaptive direct search (BADS; *Acerbi and Ma, 2017*) to optimize the model parameters. All model fits were obtained by performing several iterations of the optimization procedure with 10 different sets of starting parameters.

## Drift diffusion models of difficulty judgments

We developed four alternative models of difficulty judgments. All models assume that MCE is integrated into a DV, as in the drift diffusion model for color judgments. However, for difficulty judgments, there are two DVs: $DV_{S1}$ and $DV_{S2}$, representing the accumulated momentary evidence from the left and right stimulus, respectively. In all models, we assumed that accumulation proceeds in a serial,

time-multiplexed manner where evidence integration switches back and forth between the two stimuli (*Kang et al., 2021*). We assumed perfect time sharing between $DV_{S1}$ and $DV_{S2}$, resulting in decision times that are twice as long compared to parallel integration.

The models differ in (i) how MCE is accumulated into a decision variable and (ii) the decision criterion that is used to make the difficulty choice (*Figure 2*).

## Race model

In the race model, difficulty choice depends on a race between $DV_{S1}$ and $DV_{S2}$. The DV for each stimulus is calculated in the same way as if participants made two independent color choices (*Equation 1*) and the first decision to cross the bound is regarded as the easier decision. For difficulty choices, we did not include a color bias ($C_0$). Thus, the drift part of $DV_{S1}$ is simply $\mu_{S1} = \kappa \cdot C_{S1}^{\pm}$ (and similarly for S2).

The difficulty choice in the race model is determined by which DV crosses its bound first and the stimulus associated with this DV is chosen as the easier one. The decision time is determined by the time that the first DV crosses a bound. In the model the decision bounds collapse in the same way for both stimuli.

## Difference model

In the difference model, $DV_{S1}$ and $DV_{S2}$ are calculated in the same way as in the race model. However, the decision bound for the difficulty choice is not applied to the individual color DVs, but instead to the difference in absolute DVs, that is $|DV_{S1}| - |DV_{S2}|$. A difficulty choice is made when this difference value reaches an upper (lower) bound, indicating that S1 (S2) is the easier stimulus. The bounds collapse symmetrically, according to *Equation 2*.

## Two-step model

The two-step model is similar to the difference model, however, the bounds for difficulty choice depend on the sign of each DV, according to an initial mini-decision. The mini-decision depends on a low-threshold bound $B_{mini}$. In order to minimize the number of additional parameters in this model, we modeled $B_{mini}$ as time-independent for 2 s, followed by a sharp collapse. The DVs were calculated in the same way as in the race and difference models. Thus, choices and decision time ($t_{mini}$) for the mini-decision only depend on the signed color coherence, $C_0$ and the parameters $\kappa$ and $B_{mini}$.

The model uses the mini-decision to determine the color dominance of each stimulus and then only performs difficulty judgments assuming this color dominance. Therefore, once a mini-decision has been made for either of the two DVs, a difficulty judgment is made when $\text{sign}(DV_{S1}(t_{mini})) \cdot DV_{S1} - \text{sign}(DV_{S2}(t_{mini})) \cdot DV_{S2}$ reaches one of the bounds, that is $\pm B(t)$. We also tested a version of the two-step model in which both DVs need to reach $B_{mini}$ before the difficulty choice can be made. However, this model was inferior compared to the model presented here, in which only one mini-decision is required, and the sign of the other DV is fixed according to its value at time $t_1$ (group-level $\log_{10} \text{BF} = 49.11$ in favor of single mini-decision model).

As in the difference model, the bounds for the difficulty choice collapse over time, starting from the beginning of the accumulation process (i.e., before a mini-decision has been made). We constrained the model such that a difficulty choice could only be made at time $t_{mini}$ at earliest, that is, once a mini-decision has been made.

## Absolute momentary evidence model

In the absolute momentary evidence model, DVs represent the accumulated absolute MCE at each time step.

$$\Delta DV_t = |\mu \Delta t + \mathcal{N}(0, \sqrt{\Delta t})| \tag{3}$$

The model is, therefore, color agnostic in that it accumulates evidence in favor of strong color independent of whether the dominant color is blue or yellow on each frame. Thus, both DVs are always positive. Similar to the previous two models, a difficulty choice is made when the difference in DVs, $DV_{S1} - DV_{S2}$, exceeds an upper/lower bound, which again collapses over time.

## Model fitting

To fit each model we simulated 1000 trials for each unique combination of signed coherence for S1 × S2 (using a $\Delta t$ of 5 ms). An Epanechnikov kernel smoothed probability distribution was then fit to the simulated RT data for each choice (S1 or S2) and combination of signed coherences. This was used to calculate the log likelihood of the data given the model parameters $\kappa, u, a, d, T_{nd}$, and in case of the two-step model, $B_{mini}$. We used BADS (*Acerbi and Ma, 2017*) to optimize the model parameters. Model fits were obtained by performing several iterations of the optimization procedure with 10 different sets of starting parameters.

The BIC was computed for each model and participant in order to compare the models while controlling for their number of free parameters. Group-level model comparison was performed by summing BICs across individual participants (*Li et al., 2008*).

We also calculated the exceedance probability (*Stephan et al., 2009*; *Rigoux et al., 2014*), which measures how likely it is that any given model is more frequent than all other models in the comparison set. To do this we used the *VBA_groupBMC* command from VBA-toolbox7 (http://mbb-team. github.io/VBA-toolbox/; *Daunizeau et al., 2014*). It uses a variational Bayes method to estimate the parameters of the Dirichlet distribution that represents the probability that data from a randomly chosen participant was generated by a specific model. Models are treated as random effects that can differ across subjects. BIC values were used as an approximation to the log model evidence.

## Model recovery

We used the parameters from the fits of the race, difference, two-step, and absolute momentary evidence models to the data for each participant in Experiment 1 to generate 10 synthetic data sets for each participant. We then fit each synthetic data set with each of the four models as we did with the real data. For model recovery, we examined the proportion of times the BIC was lowest for the fit to the model that was used to generate the data. Classification accuracy was 93.5%, 97.0%, 53.5%, and 95.5% for data generated by the race, difference, two-step, and absolute momentary evidence models, respectively. For data generated by the two-step model the BIC was lowest for the difference model in 36% of fits.

## Model for Experiment 2: Difficulty judgments with unknown vs. known color dominance

In order to model choice accuracy in the unknown and known color condition, we used a difference model in which difficulty choice was based on either the difference in absolute DVs (unknown color) or appropriately signed DVs (known color). To fit choices in difficulty judgments, we excluded trials in unknown condition in which both stimuli had the same strength (i.e., $\Delta C = 0$), and trials in which the two stimuli had different dominant colors (so as to make comparable to the known color condition).

For the RT task in Experiment 2, we fit the difference model as in Experiment 1 to the data from the unknown color condition. We then used the optimized parameters to predict choices and RTs in the known color condition using the same model, but with appropriately signed DVs instead of absolute DVs.

To fit the choices in the controlled duration task we simulated 2000 trials for each unique combination of signed coherence for S1 × S2 (using a $\Delta t$ of 5 ms) with a collapsing bound as in the RT task. Choice was based on crossing a bound or on the the sign of the difference in DVs at the end of evidence accumulation if no bound had been crossed. We fit both the known and unknown color conditions simultaneously using the appropriate decision rules for each (difference model with absolute DVs for the unknown and appropriately signed DVs instead of absolute DVs for the known color condition).

We included a buffer $T_{buf}$ in the model that controls the duration of the stimulus streams that can be stored while the accumulation process alternates between the two stimuli and this was set to 80 ms based on our previous work (*Kang et al., 2021*). If the stimulus duration is shorter than the buffer, the amount of time that each stimulus is sampled, $T_{dur}$, equals the stimulus duration (i.e., all the information can be stored in the buffer and can be integrated into the DV). If the stimulus duration is longer than the buffer, $T_{dur}$ equals the buffer plus the remaining stimulus duration, time shared between the two stimuli:

$$T_{\text{dur}} = \begin{cases} T_{\text{stim}} & \text{if } T_{\text{stim}} \leq T_{\text{buf}} \\ T_{\text{buf}} + \dfrac{(T_{\text{stim}} - T_{\text{buf}})}{2} & \text{if } T_{\text{stim}} > T_{\text{buf}} \end{cases} \tag{4}$$

## Difference in confidence model

For the RT data of Experiment 2 we fit a difference in confidence model. Rather than comparing the two DVs (as in the difference model), we compare measures of confidence. Confidence is a mapping from DV and time into the probability of being correct if one were to choose color based on the sign of the DV. We followed the method in *Kiani and Shadlen, 2009*, to calculate confidence. We placed bounds on the difference in confidence (measured as the log-odds of being correct). All other elements of the model were the same as for the difference model (e.g., collapsing bounds). We fit the unknown color condition and use the fits to predict the known color condition. In the unknown color condition, we compared the difference in the absolute log-odds. In the known color condition we calculated the difference in the appropriately signed log-odds (i.e., the log-odds for choosing the known color). We chose to represent confidence as log-odds because using raw probabilities led to very poor fits. This is because for high:high coherences the difference in confidence (when both are close to 1) can be very small when represented in raw probabilities.

## Optimal RT model

We derived the decision strategy that maximizes the number of correct choices per unit time, for the RT tasks with known and unknown color dominances. To this end, we formalize the difficulty-judgment task as a POMDP. Following well-established procedures, we find the solution to the POMDP by transforming it into a fully observable MDP over belief states. The MDP comprises (*Geffner and Bonet, 2013*):

- a space $S$,
- an initial state $s_0 \in S$,
- goal states $S_G \in S$,
- a set of actions $A(s)$ applicable in each state $s \in S$,
- transition probabilities $P_a(s'|s)$ that specify the probability of transitioning to state $s'$ after selecting action $a$ in state $s$, and
- rewards and costs, $r(a, s)$, for selecting action $a$ in state $s$.

In our task, the state space is defined by the tuple $\langle t_{S1}, t_{S2}, DV_{S1}, DV_{S2} \rangle$ where $t_x$ is the time spent sampling stimulus $x$ and $DV_x$ is the accumulated evidence for stimulus $x$. We discretize time and accumulated evidence in bins of $\Delta t$ and $\Delta DV$. The initial state is the one for which no evidence has been accrued, $s_0 = \langle 0, 0, 0, 0 \rangle$.

The actions available to the decision maker are: gather more evidence, choose the option $S1$ (as the least-difficult random dot color patch), or choose the option $S2$. The last two actions terminate the trial and lead to a 'dummy' cost-free and absorbing goal state.

As with the models that we fit to the data, the two stimuli are sampled serially in rapid alternation so that we alternate between sampling and in sequential time steps ($\Delta t$).

When stimulus $Sx$ is sampled while in state $s$, the agent obtains an evidence sample which is used to update the state to $s' = \langle t'_{S1}, t'_{S2}, DV'_{S1}, DV'_{S2} \rangle$. Because one stimulus is sampled at a time, only two of the four variables that define a state can change per new observation. For example, if the stimulus $S1$ is sampled, then $t_{S2}$ and $DV_{S2}$ do not change. In turn, since time evolves in steps of $\Delta t$, then $t'_{S1} = t_{S1} + \Delta t$.

To complete the definition of the transition probabilities, it remains only to specify how the probability of $DV_x$ changes after sampling stimulus $x$, $P_{sample_x}(DV'_x|s)$. As in the models we fit to the data, we assume that the sensory observations follow a normal distribution with mean equal to $\kappa \cdot C_x^{\pm} \cdot \Delta t$ and variance equal to $\Delta t$, where $\kappa$ is a drift rate coefficient and $C_x^{\pm}$ is the signed color strength of the sampled patch. If $C_x^{\pm}$ were known, the values of $DV_x$ would follow a normal distribution:

$$P_{\text{sample}_x}(DV'_x|s, C_x^{\pm}) = \mathcal{N}(DV'_x - DV_x|\kappa \cdot C_x^{\pm} \cdot \Delta t, \Delta t), \tag{5}$$

where $DV_x$ is the state of the decision variable in state $s$, and $\mathcal{N}(\cdot|\mu, \sigma^2)$ is the normal p.d.f. with mean $\mu$ and variance $\sigma^2$. Since we do not know the color coherence $C_x^\pm$, we must marginalize over its possible values:

$$P(DV_x'|s) = \sum_{C_x^\pm} P(DV_x'|s, C_x^\pm)P(C_x^\pm|s). \tag{6}$$

For clarity of notation we omitted the associated action, $sample_x$. We assume that the decision maker knows the possible values that $C_x^\pm$ can take. The probability distribution over the color coherence values given that one is in state $s$, $P(C_x^\pm|s)$, can be calculated as (**Moreno-Bote, 2010**):

$$P(C_x^\pm|s) \propto \mathcal{N}(DV_x|\kappa \cdot C_x^\pm \cdot t_x, t_x)P(C_x^\pm), \tag{7}$$

where the constant of proportionality is such that the sum of $P(C_x^\pm|s)$ over the possible values of $C_x^\pm$ is equal to 1. The prior $P(C_x^\pm)$ is uniformly distributed over the discrete set of unsigned color coherences, as in the experiment.

We find the optimal policy by solving the Bellman equations (as in **Drugowitsch et al., 2012**). Each state $s$ has associated with it a value, $V(s)$, given by the maximum value over the actions applicable in state $s$ (in this case we have just sampled $S2$), so that

$$V(s) = \max \begin{cases} \text{choose } S1: \\ b(s, S1)R_c + \big(1 - b(s, S1)\big)(R_n - t_p\rho) - (t_{nd} + t_w)\rho, \\ \\ \text{choose } S2: \\ b(s, S2)R_c + \big(1 - b(s, S2)\big)(R_n - t_p\rho) - (t_{nd} + t_w)\rho, \\ \\ \text{sample } S1: \\ \int_{s'} P_{\text{sample } S1}(s'|s)V(s')ds' - \rho\Delta t, \end{cases} \tag{8}$$

where $b(s, Sx)$ is the probability that choosing $Sx$ is correct in state $s$, $R_c$ is the reward obtained for a correct choice, $R_n$ is the reward obtained after an incorrect choice, $t_p$ is the time penalty after an error, $t_{nd}$ is the average non-decision time, $t_w$ is the inter-trial interval (from the time a choice is registered to onset of the random dot stimuli for the next trial), and $\rho$ is the expected reward per unit of time.

The probability that choice $Sx$ is correct in state $s$, $b(s, \text{choose } Sx)$, is the probability that the color strength at stimulus $Sx$ is higher than that of the other stimulus, so that:

$$b(s, \text{choose } S1) = \sum_{C_{S1}^\pm} \sum_{C_{S2}^\pm} P(C_{S1}^\pm|s)P(C_{S2}^\pm|s)\omega(C_{S1}^\pm, C_{S2}^\pm) \tag{9}$$

where the summations are over all possible signed color coherence values, and

$$\omega(a, b) = \begin{cases} 1, & \text{if } |a| > |b|, \\ 0, & \text{if } |a| < |b|, \\ 0.5, & \text{otherwise}, \end{cases} \tag{10}$$

constrains the summation in **Equation 9** to the coherence pairs for which $S1$ is the objectively easier, plus half of the ties. Note that $b(s, \text{choose } S2) = 1 - b(s, \text{choose } S1)$.

In **Equation 8**, if $\rho$ (the reward per unit of time) were known, the Bellman equations could be solved in a single backwards pass, assuming that for sufficiently long times the best action is to select one of two terminal actions, and then propagating the value function $V(s)$ backwards in time. However, $\rho$ is not known since it depends on the decision policy itself. Following a usual procedure (**Bertsekas, 2011**), we find the value of $\rho$ by root finding. We solve the Bellman equations by backwards induction for two extreme values of $\rho$, such that the actual value lies between them. Then we divide the range into two halves, keeping the half for which the value of the initial state $V(s_0)$ has different signs for the extreme values of the range. We repeat this procedure multiple times, gradually bracketing the value

of $\rho$ in diminishing intervals, until the difference between the value we assume and the one resulting from solving Bellman's equations is negligible.

Once the value of each state converges to its optimal value, $V^*(s)$, the best action in each state is the one for which the value of the state-action pair, $Q(s,a)$, is equal to $V^*(s)$ (*Equation 8*). For the analysis shown in *Figure 8*, we simulated 200,000 trials following the optimal policy. The parameters used to make these simulations were: $\kappa = 13$, $R_c = 1$, $R_n = 0$, $t_p = 1s$, $t_{nd} = 0.4s$, $t_w = 0.5s$, $\Delta t = 0.05s$, and $\Delta DV = 0.1$.

The derivation of the optimal policy for the known color RT task follows the same procedure to the one described above for the case of unknown color, with the only exception that the signed coherences, $C^{\pm}$, is replaced by the unsigned coherences, $C$.

We did not fit the parameters of the optimal model to the data as the experiment was not designed to incentivize maximization of the reward rate and fitting would have been computationally laborious.

## Acknowledgements

This work was supported by the National Institutes of Health (R01NS117699 to DMW; R01NS113113 to MNS), and the Air Force Office of Scientific Research under award (FA9550-22-1-0337 to DMW and MNS), the Howard Hughes Medical Institute (MNS), and Kavli Institute for Brain Science (AL).

## Additional information

### Funding

| Funder | Grant reference number | Author |
|---|---|---|
| National Institutes of Health | R01NS117699 | Daniel M Wolpert |
| National Institutes of Health | R01NS113113 | Michael N Shadlen |
| Air Force Office of Scientific Research | FA9550-22-1-0337 | Michael N Shadlen |
| Howard Hughes Medical Institute | | Michael N Shadlen |
| Kavli Institute for Brain Science | | Anne Löffler |

The funders had no role in study design, data collection and interpretation, or the decision to submit the work for publication.

### Author contributions

Anne Löffler, Ariel Zylberberg, Conceptualization, Software, Formal analysis, Investigation, Methodology, Writing – original draft, Writing – review and editing; Michael N Shadlen, Conceptualization, Formal analysis, Supervision, Investigation, Methodology, Writing – original draft, Writing – review and editing; Daniel M Wolpert, Conceptualization, Resources, Software, Formal analysis, Supervision, Funding acquisition, Investigation, Methodology, Writing – original draft, Project administration, Writing – review and editing

### Author ORCIDs

Ariel Zylberberg (i) https://orcid.org/0000-0002-2572-4748
Michael N Shadlen (i) http://orcid.org/0000-0002-2002-2210
Daniel M Wolpert (i) https://orcid.org/0000-0003-2011-2790

### Ethics

Human subjects: Participants provided written informed consent prior to the study. The study was approved by the local ethics committee (IRB-AAAR9148, Institutional Review Board of Columbia University Medical Center).

Reviewer #1 (Public Review): https://doi.org/10.7554/eLife.86892.3.sa1
Reviewer #2 (Public Review): https://doi.org/10.7554/eLife.86892.3.sa2
Reviewer #3 (Public Review): https://doi.org/10.7554/eLife.86892.3.sa3
Author Response https://doi.org/10.7554/eLife.86892.3.sa4

## Additional files

### Supplementary files

• Supplementary file 1. Fit parameter values for drift diffusion model of color judgments (Experiment 1a).

• Supplementary file 2. Fit parameter values for race model (Experiment 1).

• Supplementary file 3. Fit parameter values for difference model (Experiment 1).

• Supplementary file 4. Fit parameter values for two-step model (Experiment 1).

• Supplementary file 5. Fit parameter values for absolute momentary evidence model (Experiment 1).

• Supplementary file 6. Fit parameter values for difference model in reaction time task (Experiment 2).

• Supplementary file 7. Fit parameter values for difference model in controlled duration task (Experiment 2). The high parameter for subject 2 implies that the decision process was not bounded.

• Supplementary file 8. Fit parameter values for confidence model in reaction time task (Experiment 2).

• MDAR checklist

• Source data 1. The raw Matlab data set for all the experiments.

### Data availability

The raw Matlab data set for all the experiments are in Source data 1. In addition to the data and analysis, simulation and fitting code can be found at https://doi.org/10.17632/wvkn5s479j.1.

The following dataset was generated:

| Author(s) | Year | Dataset title | Dataset URL | Database and Identifier |
|---|---|---|---|---|
| Wolpert D | 2023 | eLife difficulty data (2023) | https://doi.org/10.17632/wvkn5s479j.1 | Mendeley Data, 10.17632/wvkn5s479j.1 |

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
