## [Editor Report · eLife assessment]

This study investigates how humans make decisions on the difficulty of perceptual categorization tasks. The study finds that such judgments are best described by an evidence-accumulation model that includes a dynamic comparison of difficulty-related evidence, which terminates when the difference in evidence between two tasks reaches a predetermined bound - a **valuable** finding for research in perceptual decision-making. The paper provides **compelling** behavioral evidence for the proposed model through: 1/ quantitative model selection/validation procedures, and 2/ qualitative analyses of the relation between the optimal model of the task and the human data (and the proposed model).

---

## [Referee Report · Reviewer #1 (Public Review)]

Meta-cognition, and difficulty judgments specifically, is an important part of daily decision-making. When facing two competing tasks, individuals often need to make quick judgments on which task they should approach (whether their goal is to complete an easy or a difficult task).

In the study, subjects face two perceptual tasks on the same screen. Each task is a cloud of dots with a dominating color (yellow or blue), with a varying degree of domination - so each cloud (as a representation of a task where the subject has to judge which color is dominant) can be seen an easy or a difficult task. Observing both, the subject has to decide which one is easier.

It is well-known that choices and response times in each separate task can be described by a drift-diffusion model, where the decision maker accumulates evidence toward one of the decisions ("blue" or "yellow") over time, making a choice when the accumulated evidence reaches a predetermined bound. However, we do not know what happens when an individual has to make two such judgments at the same time, without actually making a choice, but simply deciding which task would have stronger evidence toward one of the options (so would be easier to solve).

It is clear that the degree of color dominance ("color strength" in the study's terms) of both clouds should affect the decision on which task is easier, as well as the total decision time. Experiment 1 clearly shows that color strength has a simple cumulative effect on choice: cloud 1 is more likely to be chosen if it is easier and cloud 2 is harder. Response times, however, show a more complex interactive pattern: when cloud 2 is hard, easier cloud 1 produces faster decisions. When cloud 2 is easy, easier cloud 1 produces slower decisions.

The study explores several models that explain this effect. The best-fitting model (the Difference model is the paper's terminology) assumes that the decision-maker accumulates evidence in both clouds simultaneously and makes a difficulty judgment as soon as the difference between the values of these decision variables reaches a certain threshold. Another potential model that provides a slightly worse fit to the data is a two-step model. First, the decision maker evaluates the dominant color of each cloud, then judges the difficulty based on this information.

Importantly, the study explores an optimal model based on the Markov decision processes approach. This model shows a very similar qualitative pattern in RT predictions but is too complex to fit to the real data. Possibly, the fact that simple approaches such as the Difference model fit the data best could suggest the existence of some cognitive constraints that play a role in difficulty judgments and could be explored in future research.

The Difference model produces a well-defined qualitative prediction: if the dominant color of both clouds is known to the decision maker, the overall RT effect (hard-hard trials are slower than easy-easy trials) should disappear. Essentially, that turns the model into the second stage of the two-stage model, where the decision maker learns the dominant colors first. The data from Experiment 2 impressively confirms that prediction and provides a good demonstration of how the model can explain the data out-of-sample with a predicted change in context.

Overall, the study provides a very coherent and clean set of predictions and analyses that advance our understanding of meta-cognition. The field would benefit from further exploration of differences between the models presented and new competing predictions (for instance, exploring how the sequential presentation of stimuli or attentional behavior can impact such judgments). Finally, the study provides a solid foundation for future neuroimaging investigations.

---

## [Referee Report · Reviewer #2 (Public Review)]

Starting from the observation that difficulty estimation lies at the core of human cognition, the authors acknowledge that despite extensive work focusing on the computational mechanisms of decision-making, little is known about how subjective judgments of task difficulty are made. Instantiating the question with a perceptual decision-making task, the authors found that how humans pick the easiest of two stimuli, and how quickly these difficulty judgments are made, are best described by a simple evidence accumulation model. In this model, perceptual evidence of concurrent stimuli is accumulated and difficulty is determined by the difference between the absolute values of decision variables corresponding to each stimulus, combined with a threshold crossing mechanism. Altogether, these results strengthen the success of evidence accumulation models in describing human decision-making, now extending it to judgments of difficulty.

The manuscript addresses a timely question and is very well written, with its goals, methods and findings clearly explained and directly relating to each other. The authors are specialists of evidence accumulation tasks and models. Their modelling of human behaviour within this framework is state-of-the-art. In particular, their model comparison is guided by qualitative signatures which are diagnostic to tease apart different models (e.g., the RT criss-cross pattern). Human behaviour is then inspected for these signatures, instead of relying exclusively on quantitative comparison of goodness-of-fit metrics.

The study has potential limitations well flagged by the authors after the revision process. The main limitation pertains to the (dis)similarity between the behavioural task used in the study and difficulty judgments people actually do in real world (and which are well illustrated in the introduction). First, difficulty judgments made in the task never impact the participant (a new trial simply follows) while difficulty judgments in the wild often determine whether to pursue or quit the corresponding task, which can have consequences years after the difficulty estimation (e.g., deciding to engage in a particular academic path as a function of the estimated difficulty). Second, while trial-by-trial feedback is delivered in the task, difficulty estimation in the wild has to be made with partial information and feedback is either absent or delayed. How much these differences are key in providing an accurate computational description of human difficulty judgments will likely require further research.

Another limitation is the absence of models based on computational principles other than evidence accumulation. Although there are good reasons to favour evidence accumulation models in these settings (as mentioned by the authors in their manuscript), showing that evidence accumulation models would have won against competitors would have further strengthened the authors' claim that difficulty judgment about perceptual information are firmly anchored in the principles of evidence accumulation.

These limitations should not distract the reader from the impact of the present work, which will likely be wide, spanning the whole field of decision-making, and this across species. It will echo in particular with the many other seminal studies that have relied on a similar theoretical account of behaviour and brain activity (evidence accumulation). In addition, this study will hopefully inspire novel task designs aiming at addressing difficulty judgment estimations in controlled lab experiments, possibly with features closer to real world difficulty estimation (e.g., long-term consequences of difficulty estimation and absence of feedback).

---

## [Referee Report · Reviewer #3 (Public Review)]

The manuscript presents novel findings regarding the judgment of difficulty of perceptual decisions. In the main task (Experiment 1), participants accumulated evidence over time about two tasks, patches of random dot motion, and were asked to report for which patch it would be easier to make a decision about its dominant color, while not explicitly making such decision(s). By fitting several alternative models, authors demonstrated that while accuracy changes as a function of the difference between stimulus strengths, reaction times of such decisions are not solely governed by the difference in stimulus strength, but (also) by the difference in absolute accumulated evidence for color judgment of the two stimuli ('Difference model'). Predictions from the best fitted model were then tested with a new set of conditions and participants (Experiment 2). Here, authors eliminated part of the uncertainty by informing participants about the dominant color of the two stimuli ('known color' condition) and showing that reaction times were faster compared to the 'unknown color' task, and only depended on the difference between stimulus strengths.

The paper deals with a valuable question about a metacognitive aspect of perceptual decision making, which was only sparsely addressed before. The paper is very well written, figures and illustrations clearly accompanied the text, and methods and modeling are rigor. The authors also address the concern that a difficulty judgment might be a confidence estimation, another metacognitive judgment of perceptual decisions, by fitting a Confidence model to the 'known color' condition in Experiment 2 and showing that this model performs worse compared to the Difference model. This is an important control analysis, given the possibility that humans might make an implicit decision about the dominant color of each patch, and then report their level of confidence.

This work is likely to be of great interest in the field of behavioral modeling of perceptual decision making, and might encourage further investigations of how judging the difficulty of a task affects subsequent decisions about the same task.

---

## [Author Response]

The following is the authors’ response to the original reviews.

**Reviewer #1 (Public Review):**
Meta-cognition, and difficulty judgments specifically, is an important part of daily decision-making. When facing two competing tasks, individuals often need to make quick judgments on which task they should approach (whether their goal is to complete an easy or a difficult task).In the study, subjects face two perceptual tasks on the same screen. Each task is a cloud of dots with a dominating color (yellow or blue), with a varying degree of domination - so each cloud (as a representation of a task where the subject has to judge which color is dominant) can be seen an easy or a difficult task. Observing both, the subject has to decide which one is easier.It is well-known that choices and response times in each separate task can be described by a driftdiffusion model, where the decision maker accumulates evidence toward one of the decisions (”blue” or ”yellow”) over time, making a choice when the accumulated evidence reaches a predetermined bound. However, we do not know what happens when an individual has to make two such judgments at the same time, without actually making a choice, but simply deciding which task would have stronger evidence toward one of the options (so would be easier to solve).It is clear that the degree of color dominance (”color strength” in the study’s terms) of both clouds should affect the decision on which task is easier, as well as the total decision time. Experiment 1 clearly shows that color strength has a simple cumulative effect on choice: cloud 1 is more likely to be chosen if it is easier and cloud 2 is harder. Response times, however, show a more complex interactive pattern: when cloud 2 is hard, easier cloud 1 produces faster decisions. When cloud 2is easy, easier cloud 1 produces slower decisions.The study explores several models that explain this effect. The best-fitting model (the Difference model is the paper’s terminology) assumes that the decision-maker accumulates evidence in both clouds simultaneously and makes a difficulty judgment as soon as the difference between the values of these decision variables reaches a certain threshold. Another potential model that provides a slightly worse fit to the data is a two-step model. First, the decision maker evaluates the dominant color of each cloud, then judges the difficulty based on this information.

Thank you for a very good summary of our work.

Importantly, the study explores an optimal model based on the Markov decision processes approach. This model shows a very similar qualitative pattern in RT predictions but is too complex to fit to the real data. It is hard to judge from the results of the study how the models identified above are specifically related to the optimal model - possibly, the fact that simple approaches such as the Difference model fit the data best could suggest the existence of some cognitive constraints that play a role in difficulty judgments.

The reviewer asks “how the models identified above are specifically related to the optimal model”. We did fit the four models to simulations of the optimal model and found that the Difference model was the closest. However, we did not fit the parameters of the optimal model to the data (no easy feat given the complexity of the model) as the experiment was not designed to incentivize maximization of the reward rate and fitting would have been computationally laborious. We therefore focused on the qualitative features of the optimal model and how they compare to our models. We now also include the optimal model for the known color dominance RT experiment (line 420). We have also added a new paragraph in the Discussion on the optimal model at line 503 comparing it qualitatively to the Difference model.

The Difference model produces a well-defined qualitative prediction: if the dominant color of both clouds is known to the decision maker, the overall RT effect (hard-hard trials are slower than easyeasy trials) should disappear. Essentially, that turns the model into the second stage of the twostage model, where the decision maker learns the dominant colors first. The data from Experiment 2 impressively confirms that prediction and provides a good demonstration of how the model can explain the data out-of-sample with a predicted change in context.Overall, the study provides a very coherent and clean set of predictions and analyses that advance our understanding of meta-cognition. The field would benefit from further exploration of differences between the models presented and new competing predictions (for instance, exploring how the sequential presentation of stimuli or attentional behavior can impact such judgments). Finally, the study provides a solid foundation for future neuroimaging investigations.

Thank you for your positive comments and suggestions.

**Reviewer #2 (Public Review):**
Starting from the observation that difficulty estimation lies at the core of human cognition, the authors acknowledge that despite extensive work focusing on the computational mechanisms of decision-making, little is known about how subjective judgments of task difficulty are made. Instantiating the question with a perceptual decision-making task, the authors found that how humans pick the easiest of two stimuli, and how quickly these difficulty judgments are made, are best described by a simple evidence accumulation model. In this model, perceptual evidence of concurrent stimuli is accumulated and difficulty is determined by the difference between the absolute values of decision variables corresponding to each stimulus, combined with a threshold crossing mechanism. Altogether, these results strengthen the success of evidence accumulation models, and more broadly sequential sampling models, in describing human decision-making, now extending it to judgments of difficulty.The manuscript addresses a timely question and is very well written, with its goals, methods and findings clearly explained and directly relating to each other. The authors are specialists in evidence accumulation tasks and models. Their modelling of human behaviour within this framework is state-of-the-art. In particular, their model comparison is guided by qualitative signatures which are diagnostic to tease apart the different models (e.g., the RT criss-cross pattern). Human behaviour is then inspected for these signatures, instead of relying exclusively on quantitative comparison of goodness-of-fit metrics. This work will likely have a wide impact in the field of decisionmaking, and this across species. It will echo in particular with many other studies relying on the similar theoretical account of behaviour (evidence accumulation).

Thank you for these generous comments.

A few points nevertheless came to my attention while reading the manuscript, which the authors might find useful to answer or address in a new version of their manuscript.1. The authors acknowledge that difficulty estimation occurs notably before exploration (e.g., attempting a new recipe) or learning (e.g., learning a new musical piece) situations. Motivated by the fact that naturalistic tasks make difficult the identification of the inference process underlying difficulty judgments, the authors instead chose a simple perceptual decision-making task to address their question. While I generally agree with the authors’s general diagnostic, I am nevertheless concerned so as to whether the task really captures the cognitive process of interest as described in the introduction. As coined by the authors themselves, the main function of prospective difficulty judgment is to select a task which will then ultimately be performed, or reject one which won’t. However, in the task presented here, participants are asked to produce difficulty judgments without those judgements actually impacting the future in the task. A feature thus key to difficulty judgments thus seems lacking from the task. Furthermore, the trial-by-trial feedback provided to participants also likely differ from difficulty judgments made in real world. This comment is probably difficult to address but it might generally be useful to discuss the limitations of the task, in particular in probing the desired cognitive process as described in introduction. Currently, no limitations are discussed.

We have added a Limitations paragraph to the Discussion and one item we deal with is the generalization of the model to more complex tasks (line 539).

1. The authors take their findings as the general indication that humans rely on accumulation evidence mechanisms to probe the difficulty of perceptual decisions. I would probably have been slightly more cautious in excluding alternative explanations. First, only accumulation models are compared. It is thus simply not possible to reach a different conclusion. Second, even though it is particularly compelling to see untested predictions from the winning model in experiment #1 to be directly tested, and validated in a second experiment, that second experiment presents data from only 3 participants (1 of which has slightly different behaviour than the 2 others), thereby limiting the generality of the findings. Third, the winning model in experiment #1 (difference model) is the preferred model on 12 participants, out of the 20 tested ones. Fourth, the raw BIC values are compared against each other in absolute terms without relying on significance testing of the differences in model frequency within the sample of participants (e.g., using exceedance probabilities; see Stephan et al., 2009 and Rigoux et al., 2014). Based on these different observations, I would thus have interpreted the results of the study with a bit more caution and avoided concluding too widely about the generality of the findings.

Thank you for these suggestions.

i) We have now make it clear in the Results (line 126) that all four models we examine are accumu-lation models. In addition, we have added a paragraph on Limitations (line 530) in the Discussion where we explain why we only consider accumulation models and acknowledge that there are other non-accumulation models.

ii) Each of three participants in Experiment 2 performed 18 sessions making it a large and valuabledataset necessary to test our hypothesis. We have now included a mention of the the small number of participants in Experiment 2 in a Limitations paragraph in the Discussion (line 539).

iii) As suggested, we have now calculated exceedance probabilities for the 4 models which gives[0,0.97,0.03,0]. This shows that there is a 0.97 probability of the Difference model being the most frequent and only a 0.03 probability of the two-step model. We have included this in the results on line 237.

1. Deriving and describing the optimal model of the task was particularly appreciated. It was however a bit disappointing not to see how well the optimal model explains participants behaviour and whether it does so better than the other considered models. Also, it would have been helpful to see how close each of the 4 models compared in Figures 2 & 3 get to the optimal solution. Note however that neither of these comments are needed to support the authors’ claims.

The reviewer asks how close each of the four models is to the optimal solution. We did fit the four models to simulations of the optimal model and found that the Difference model was the closest. However, we did not fit the parameters of the optimal model to the data (no easy feat given the complexity of the model) as the experiment was not designed to incentivize maximization of the reward rate and fitting would have been computationally laborious. We therefore focused on the qualitative features of the optimal model and how they compare to our models. We now also include the optimal model for the known color dominance RT experiment (line 420). We have also added a new paragraph in the Discussion on the optimal model at line 503 comparing it qualitatively to the Difference model.

1. The authors compared the difficulty vs. color judgment conditions to conclude that the accumulation process subtending difficulty judgements is partly distinct from the accumulation process leading to perceptual decisions themselves. To do so, they directly compared reaction times obtained in these two conditions (e.g. ”in other cases, the two perceptual decisions are almost certainly completed before the difficulty decision”). However, I find it difficult to directly compare the ’color’ and ’difficulty’ conditions as the latter entails a single stimulus while the former comprises two stimuli. Any reaction-time difference between conditions could thus I believe only follow from asymmetric perceptual/cognitive load between conditions (at least in the sense RT-color < RT-difficulty). One alternative could have been to present two stimuli in the ’color’ condition as well, and asking participants to judge both (or probe which to judge later in the trial). Implementing this now would however require to run a whole new experiment which is likely too demanding. Perhaps the authors could instead also acknowledge that this a critical difference between their conditions, which makes direct comparison difficult.

We feel we can rule out that participants make color decisions (as in the color task) to make difficulty decisions. For example, making a color choice for 0% color strength takes longer than a difficulty choice for 0:52% color strengths. Thus, the difficulty judgment does not require completion of the color decisions. Therefore, average reaction time for a single color patch (C*S*11) can be longer than the reaction time for the difficulty task which contains the same coherence (C*S*11) for one of the patches. This is true despite the difficulty decision requiring monitoring of two patches(which might be expected to be slower than monitoring one patch). We have added this in to the Discussion at line 449.

**Reviewer #3 (Public Review):**
The manuscript presents novel findings regarding the metacognitive judgment of difficulty of perceptual decisions. In the main task, subjects accumulated evidence over time about two patches of random dot motion, and were asked to report for which patch it would be easier to make a decision about its dominant color, while not explicitly making such decision(s). Using 4 models of difficulty decisions, the authors demonstrate that the reaction time of these decisions are not solely governed by the difference in difficulties between patches (i.e., difference in stimulus strength), but (also) by the difference in absolute accumulated evidence for color judgment of the two stimuli. In an additional experiment, the authors eliminated part of the uncertainty by informing participants about the dominant color of the two stimuli. In this case, reaction times were faster compared to the original task, and only depended on the difference between stimulus strength.Overall, the paper is very well written, figures and illustrations clearly and adequately accompanied the text, and the method and modeling are rigor.The weakness of the paper is that it does not provide sufficient evidence to rule out the possibility that judging the difficulty of a decision may actually be comparing between levels of confidence about the dominant color of each stimulus. One may claim that an observer makes an implicit color decision about each stimulus, and then compares the confidence levels about the correctness of the decisions. This concern is reflected in the paper in several ways:

We tested a Difference in confidence model (line 315) in the orginal paper and showed it was inferior to the Difference model. We did this for experiment 2, RT task so that we could fit the unknown color condition and try to predict the known color condition. To emphasize this model (which we think the reviewer may have missed) we have moved the supplementary figure to the main results (now Fig. 6) as we think it is very cool that we were able to discard the confidence model.

When comparing the confidence model to the Difference we found the difference model was pre-Δ ferred with BIC of 38, 56, 47. We are unsure why the reviewer feels this “does not provide sufficient evidence to rule out the possibility that judging the difficulty of a decision may actually be comparing between levels of confidence about the dominant color of each stimulus.” We regard this as strong evidence.

1. It is not clear what were the actual instructors to the participants, as two different phrasings appear in the methods: one instructs participants to indicate which stimulus is the easier one and the other instructs them to indicate the patch with the stronger color dominance. If both instructions are the same, it can be assumed that knowing the dominant color of each patch is in fact solving the task, and no judgment of difficulty needs to be made (perhaps a confidence estimation). Since this is not a classical perceptual task where subjects need to address a certain feature of the stimuli, but rather to judge their difficulties, it is important to make it clear.

We now include the precise words used to instruct the participant (line 604): “Your task is to judge which patch has a stronger majority of yellow or blue dots. In other words: For which patch do you find it easier to decide what the dominant color is? It does not matter what the dominant color of the easier patch is (i.e., whether it is yellow or blue). All that matters is whether the left or right patch is easier to decide”.

Knowing both colors or the dominant color is not sufficient to solve the task. Knowing both are yellow does not tell you which has more yellow which is what you need to estimate to solve the task. Again, we tested a confidence model in the original version of the paper and showed it was a poor model compared to the Difference model.

1. Two step model: two issues are a bit puzzling in this model. First, if an observer reaches a decision about the dominant color of each patch, does it mean one has made a color decision about the patches? If so, why should more evidence be accumulated? This may also support the possibility that this is a ”post decision” confidence judgment rather than a ”pre decision” difficulty judgment. Second, the authors assume the time it takes to reach a decision about the dominant color for both patches are equal, i.e., the boundaries for the ”mini decision” are symmetrical. However, it would make sense to assume that patches with lower strength would require a longer time to reach the boundaries.

In the Two-step model we assume a mini decision is made for the color of each stimulus. However, the assumption is that this is made with a low bound so it is not a full decision as in a typical color decision. Again estimating the colors from the mini decision does not tell you which is easier so you need to accumulate more evidence to make this judgment. In fact the Race model is a version of the two step in which no further accumulation is made after the initial decision and this model fits poorly (we now explain this on line 185). We assume for simplicity that the first stimulus to cross a bound triggers both mini color decisions. So although the bounds are equal the one with stronger color dominance is more likely to hit the bound first.

We have already addressed this concern about the comparison with confidence above.

1. Experiment 2: the modification of the Difference model to fit the known condition (Figure 5b),can also be conceptualized as the two-step model, excluding the ”mini” color decision time. These two models (Difference model with known color; two-step model) only differ from each other in a way that in the former the color is known in advance, and in the second, the subject has to infer it. One may wonder if the difference in patterns between the two (Figure 3C vs. Figure 6B) is only due to the inaccuracies of inferring the dominant color in the two-step model.

In Experiment 2 the participant is explicitly informed as to the color dominance of both stimuli. Therefore, assuming the two-step model skips the first step and uses this explicit information in the second step, the difference and two-step model are identical for modeling Experiment 2. We explain this now on line 277.

As the reviewer suggests, differences in predictions between the Difference and Two-step arise from trials in which there is a mismatch between the inferred dominant colors from the two-step model and the color associated with the final DVs in the Difference model. We now explain this on line 187. We do not see this as a problem of any sort but just defines the difference between the models. Note that the new exceedance analysis now strongly supports the Difference model as the most common model among the participants.

An additional concern is about the controlled duration task: Why were these specific durations chosen (0.1-1.65 sec; only a single duration was larger than 1sec), given the much longer reaction times in the main task (Experiment 1), which were all larger on average than 1sec? This seems a bit like an odd choice. Additionally, difficulty decision accuracies in this version of the task differ between known and unknown conditions (Figure 7), while in the reaction time version of the same task there were no detectable differences in performance between known and unknown conditions (Figure 6C), just in the reaction times. This discrepancy is not sufficiently explained in the manuscript. Could this be explained by the short trial durations?

The reviewer asks about the choice of stimulus durations in Experiment 2. First, RTs in Experiment 1 do not only reflect the time needed to make decisions but also contain non-decision times (0.23-0.47 s). So to compare decision time in RT and controlled duration experiment one must subtract the non-decision time from the RTs (the non-decision time is not relevant to the controlled duration experiment). Second, the model specifically predicts that differences in performance between the known and unknown color dominance conditions are largest for short duration stimulus presentation trials (see Fig. 7). We explain this on line 346. For long durations, performance pretty much plateaus, and many decisions have already terminated (Kiani 2008). We sample stimulus durations from a discrete truncated exponential distribution to get roughly equal changes in accuracy between consecutive durations (which we now explain at line 345).

Group consensus reviewThe reviewers have discussed with each other, and they have discussed a series of revisions which, if carried out, would make their evaluation of your paper even more positive. I outline them below in case you would be interested in revising your paper based on these reviews. You will see below that the reviewers share overall a quite positive evaluation of your study. All three limitations described in the Public Reviews could be addressed explicitly in the discussion which for the moment is limited to description and generalization of findings.1. The model selection procedure should be amended and strengthened to provide clearer results. As noted by one of the reviewers during the consultation session, ”the Difference model just barely wins over the two-step model, and the two-step model might produce the same prediction for the next experiment.” You will also see below that Reviewer #2 provides guidance to improve the model selection process: ”[...] the second experiment presents data from only 3 participants (1 of which has slightly different behaviour than the 2 others), thereby limiting the generality of the findings. Third, the winning model in experiment #1 (difference model) is the preferred model on 12 participants, out of the 20 tested ones. Fourth, the raw BIC values are compared against each other in absolute terms without relying on significance testing of the differences in model frequency within the sample of participants (e.g., using exceedance probabilities; see Stephan et al., 2009 and Rigoux et al., 2014).” Altogether, model selection appears currently to be the ’weakest’ part of the paper (Difference model vs. Two-step model, model comparison, how to better incorporate the optional model with the other parts). It would be great if you would improve this section of the Results.

Thank you for these suggestions.

i) We have now make it clear in the Results (line 126) that all four models we examine are accumu-lation models. In addition, we have added a paragraph on Limitations (line 530) in the Discussion where we explain why we only consider accumulation models and acknowledge that there are other non-accumulation models.

ii) Each of three participants in Experiment 2 performed 18 session making it a large and valuabledataset necessary to test our hypothesis. We have now included a mention of the the small number of participants in Experiment 2 in a Limitations paragraph in the Discussion (line 539).

iii) We have now calculated exceedance probabilities for the 4 models which gave [0,0.97,0.03,0].This shows that there is a 0.97 probability of the Difference model being the most frequent and only a 0.03 probability of the two-step model. We have included this in the results on line 237.

1. All reviewers have noted that the relation of the optimal model with the human data and theother models should be clarified and discussed in a revised version of the manuscript. You will find their specific comments in their individual reviews, appended below.

We now include the optimal model for the known color dominance RT experiment (line 420). We have also added a new paragraph in the Discussion on the optimal model at line 503 comparing it to the Difference model.

1. Finally, the exclusion strategy is also unclear at the moment and should be clarified and discussed explicitly somewhere in a revised version of the manuscript. Reviewers were wondering why so many participants were excluded from Experiment 1, and only 3 participants were included in Experiment 2. This should also be clarified better in the manuscript.

We have clarified the exclusion criteria in the Methods at line 651 as a new subsection.

The data quality problem with MTurk is well documented (Chmielewski, M & Kucker SC. 2020. An MTurk Crisis? Shifts in Data Quality and the Impact on Study Results. Social Psychological and Personality Science, 11, 464-473). Given that this was an online experiment on MTurk, it is hard to know exactly why some participants showed low accuracy, but it’s likely that some may have misunderstood the instructions in the difficulty task or they may have been unmotivated to do well in this highly repetitive task. Either reason would be problematic for our model comparisons that are based on choice-RT patterns. Note that the cut-offs we chose for inclusion were purely based on accuracy, whereas the modeling approach considered RTs, which importantly were not used as a inclusion criterion (see revised methods). Moreover, accuracy cut-offs were fairly lenient and mainly aimed to exclude participants who appeared to be guessing/misunderstood instructions (for reference: mean sensitivity of participants who were included was 2x higher than the cut-offs we used).

Each of three participants in Experiment 2 performed 18 session making it a large and valuable dataset necessary to test our hypothesis. We have now included a mention of the the small number of participants in Experiment 2 in a Limitations paragraph in the Discussion (line 539).

**Reviewer #1 (Recommendations For The Authors):**
Thank you for an excellent paper, I enjoyed reading it a lot. I have a few questions that could potentially clarify some aspects for the reader.(1) It seems from the model fit plots (Figure 3) that the RT predictions of the model tend to overshoot in cases where one of the clouds is very easy. Could you include potential interpretations of this effect?

We assume the reviewer is examining the Difference Model (i.e. the preferred model) panel when commenting on the overshoot. It is true the predictions for the highest coherence (bottom purple line) for RT is above the data but it is barely outside the data errorbars of 1 s.e. To be honest we regard this as a pretty good fit and would not want to over-interpret this small mismatch.

(2) On page 4, around line 121, the study discusses the ”criss-crossing” effect in the RT data. You mention that the fact that RTs are long in hard-hard trials compared to easy-easy trials could be important here: ”These tendencies lead to a criss-cross pattern..”. It is confusing since, for instance, the race model does not have a criss-cross, but still exhibits the overall effect. I was intrigued bythe criss-crossing, and after some quick simulations, I found that the equation RT2 ∗ = 2 − 2 ∗ Cs12 −Cs22 + 6 ∗ (Cs1 ∗ Cs2)2 can (very roughly) replicate Figure 1d (bottom panel), so it seems that the criss-crossing effect must be produced by some interactive effect of color strengths on RTs. I wonder if you could provide a better explanation of how this interactive effect is generated by the model, given that it is the main interesting finding in the data. I believe at this point the intuition is not well-outlined.

The criss cross arises through an interaction of the coherences as the reviewer suspects. That is, for the Difference model the RT related to abs(|Coh1|- |Coh2|). If we replace the first abs with a square we get

|coh1|2 + |coh2|2 − 2|coh1||coh2|

The larger this is, the smaller the RT so

RT = constant − coh12 − coh22 + 2|coh1||coh2|

which is very similar to the formula the reviewer mentions.

We now supply an intuition as to why the criss-cross arises in the Difference model (line 167). We do not get a criss-cross in the race model, because there the RT is determined by the Race that that reaches a bound first. Because the races are independent, RTs will be fastest when coherence is high for either stimuli.

(3) Am I wrong in my intuition that the two-step model would produce very similar predictions as the Difference model for Experiment 2? It would be great to discuss that either way since the twostep model seems to produce very close quantitative and pretty much the same qualitative fit to the data of Experiment 1.

In Experiment 2 the participant is explicitly informed about the color dominance of both stimuli. Therefore, assuming the two-step model skips the first step and uses this explicit information in the second step, the difference and two-step model are identical for modeling Experiment 2. We explain this now on line 277.

(4) The inclusion of the optimal model is great. It would be beneficial to provide some more connections to the rest of the paper here. Would this model produce similar predictions for Experiment 2, for instance?

We now include the optimal model for the known color dominance RT experiment (line 420). We have also added a new paragraph in the Discussion on the optimal model at line 503 comparing it to the Difference model.

(5) In the Methods, it is quite striking that out of 51 original participants, most were excluded and only 20 were studied. It is not easy to trace through this section why and how and who was excluded, so it would be great if this information was organized and presented more clearly.

We have clarified this in the Methods at line 651 as a new subsection in the Methods. We also explain that exclusion was not made on RT data which is our main focus in the models.

**Reviewer #2 (Recommendations For The Authors):**
As detailed in the ’public review’, a more cautious discussion, notably delineating the limitations of the study would be appreciated.In their models, the authors assume that participants sequentially allocate attention between the two stimuli, alternating between them. Did the authors test this assumption and did they consider the possibility that participants could sample from both stimuli in parallel? In particular, does the conclusion of the model comparison also holds under this parallel processing assumption?

Our results are not affected by whether participants sample the stimulus sequentially through alternation or in a parallel manner (Kang et al., 2021). What does change is the parameters of the model (but not their predictions/fits). In the parallel model, information is acquired at twice the rate of the serial model. We can, therefore, obtain the parameters of parallel models (that has identical predictions to the serial model) directly from the parameters of the current sequential models simply by adjusting the parameters that depend on the time scale (subscripts s and p for serial and parallel models): κp=κs/2, up=us2, ap=as/2, and dp=2ds (Equation 2). We now explainthis on line 518.

I found the small paragraph corresponding to lines 193-196 particularly difficult to understand. If the authors could think of a better way to phrase their claim, it would probably help.

We have rewritten this paragraph at line 211

I found a type on line 122: ”wheres” instead of ”whereas”.

Corrected

I found a type on line 181: ”or” instead of ”of”.

Yes corrected

Figure #2 is extremely useful in understanding the models and their differences, make sure it remains after addressing the reviews!

Thank you, this figure is retained.

**Reviewer #3 (Recommendations For The Authors):**
All comments are detailed in the public review, with some clarifications here:1. The confusing instructions to the participants are detailed here: under ”overview of experimental tasks” in the methods it says: ”They were instructed... to indicate whether the left or right stimulus was the easier one” (line 520), and below it ”they were required to indicate which patch had the stronger color dominance...” (line 524).

We have clarified the instructions by providing the actual text displayed to participants in the methods and have ensured consistency in the method to talk about judging the easier stimulus(line 604).

The instructions were “Your task is to judge which patch has a stronger majority of yellow or blue dots. In other words: For which patch do you find it easier to decide what the dominant color is?It does not matter what the dominant color of the easier patch is (i.e., whether it is yellow or blue). All that matters is whether the left or right patch is easier to decide”.

1. Minor comments: Line 76: ”that” should be ”than”.

Thanks, corrected

Line 574: ”variable duration task” means ”controlled duration task”?

Yes, corrected

Line 151: ”or” should be ”of”.

Corrected